behaviour, ecology

carry-over effects, boldness, life-history trade-offs, pace-of-life syndrome, reversible state effects, annual cycle

**Author for correspondence:**
Stephanie M. Harris
e-mail: stephh@liv.ac.uk

# Personality-specific carry-over effects on breeding

Stephanie M. Harris[1,2], Sébastien Descamps[3], Lynne U. Sneddon[4], Milena Cairo[2], Philip Bertrand[3,5] and Samantha C. Patrick[2]

[1]Cornell Lab of Ornithology, Cornell University, 159 Sapsucker Woods Road, Ithaca, USA
[2]School of Environmental Sciences, University of Liverpool, Liverpool, UK
[3]Norwegian Polar Institute, Fram Centre, Tromsø, Norway
[4]Department of Biological and Environmental Sciences, University of Gothenburg, Gothenburg, Sweden
[5]Department of Biology and Centre for Northern Studies, Université du Québec à Rimouski, Canada

SMH, 0000-0002-8580-9444; SD, 0000-0003-0590-9013; LUS, 0000-0001-9787-3948; PB, 0000-0003-4519-6556; SCP, 0000-0003-4498-944X

Carry-over effects describe the phenomenon whereby an animal's previous conditions influence its subsequent performance. Carry-over effects are unlikely to affect individuals uniformly, but the factors modulating their strength are poorly known. Variation in the strength of carry-over effects may reflect individual differences in pace-of-life: slow-paced, shyly behaved individuals are thought to favour an allocation to self-maintenance over current reproduction, compared to their fast-paced, boldly behaved conspecifics (the pace-of-life syndrome hypothesis). Therefore, detectable carry-over effects on breeding should be weaker in bolder individuals, as they should maintain an allocation to reproduction irrespective of previous conditions, while shy individuals should experience stronger carry-over effects. We tested this prediction in black-legged kittiwakes breeding in Svalbard. Using miniature biologging devices, we measured non-breeding activity of kittiwakes and monitored their subsequent breeding performance. We report a number of negative carry-over effects of non-breeding activity on breeding, which were generally stronger in shyer individuals: more active winters were followed by later breeding phenology and poorer breeding performance in shy birds, but these effects were weaker or undetected in bolder individuals. Our study quantifies individual variability in the strength of carry-over effects on breeding and provides a mechanism explaining widespread differences in individual reproductive success.

## 1. Introduction

A fundamental challenge in ecology is understanding why individuals vary in breeding performance. An animal's previous history can be a major determinant of its fitness later in life, a phenomenon referred to as carry-over effects [1–4]. In particular, events and processes that occur prior to the current breeding season (e.g. during the non-breeding season or in previous breeding seasons) can carry over to impact future breeding success [1]. For example, studies have demonstrated that factors such as food availability [5,6], hormone levels [7,8], habitat use [9,10] and foraging behaviour [11,12] outside the breeding season can all influence subsequent reproduction. Individuals can differ in how they respond to conditions [13], and therefore intrinsic variation is thought to be important [14]. However, the sources of individual variation in carry-over effects remain poorly understood.

Carry-over effects result from life-history trade-offs among competing functions [2,15], but are rarely framed as such. When energetic reserves are limited, high allocation to current reproduction reduces potential allocation to somatic maintenance, future breeding and survival, and so animals may divert resources away from current breeding towards other functions [16–18]. Examining

carry-over effects in the framework of life-history trade-offs may offer new insights into the intrinsic factors which shape them. This is because the trade-off between current versus future reproduction also manifests in the form of different life-history strategies at the individual level. Life-history strategies are thought to occur along a fast-slow pace-of-life continuum, whereby a fast pace-of-life is characterized by high allocation to current breeding but low survival [17,19,20]. It may then be predicted that individual differences in pace-of-life should be reflected in the strength of carry-over effects on current breeding, with stronger effects of previous conditions on breeding in slow-paced than in fast-paced animals.

Among individuals, variation in pace-of-life is thought to be linked to phenotypic differences in behavioural traits, or animal personalities (the pace-of-life syndrome hypothesis; [21]). Individuals adopting a slow pace-of-life should minimize risk-taking behaviours to favour survival probability, while fast-paced individuals should adopt risky (or 'bold') behaviours that facilitate current reproduction [21,22]. Boldness should therefore predict variation in carry-over effects. While challenging non-breeding conditions should result in reduced allocation to reproduction in shy, slow-paced individuals, boldly behaved, fast-paced individuals should maintain allocation to reproduction, such that carry-over effects are weaker or undetected.

Here, we investigate personality as a predictor of carry-over effects on breeding in a species of seabird, the black-legged kittiwake (*Rissa tridactyla*). A previous study has demonstrated that pace-of-life can shape allocation trade-offs in two kittiwake populations differing markedly in pace-of-life. Following experimentally induced stress, birds from the fast-paced population maintained provisioning rates and successfully reared offspring, whereas slow-paced individuals reduced parental care, resulting in decreased offspring survival [23]. While populations of the same species are often shown to vary in pace-of-life, probably driven by their evolution under different ecological conditions [24], empirical examination of the pace-of-life syndrome at the individual level has yielded mixed results, despite theoretical support for its existence [21,25]. An increasing body of evidence demonstrates the individual variation in allocation trade-offs in the form of naturally occurring carry-over effects between seasons, with profound consequences for individual fitness [1]. However, to our knowledge, no study has previously tested whether differences in carry-over effects can be explained by individual variation in pace-of-life.

While most sources of variation in carry-over effects are poorly known, sex-dependent carry-over effects have been reported in a number of systems [26–28]. Sex-dependent carry-over effects can arise because of sex differences in breeding roles. For instance, a number of studies on birds have reported that carry-over effects on breeding phenology are stronger in females than in males, potentially owing to greater control over the timing of egg laying by females [26–29]. Sex differences in pace-of-life may also generate variation in carry-over effects: owing fundamentally to gamete dimorphism (anisogamy), males are generally expected to exhibit a faster pace-of-life relative to females, allocating towards reproductive output over longevity [30–33]. As a result, females may be subject to stronger carry-over effects on breeding, even in species where the sexes do not differ greatly in breeding roles (e.g. [29]). We therefore also examined sex differences in carry-over effects.

We examined carry-over effects on breeding using a long-term biologging dataset on kittiwakes breeding in Svalbard.

Kittiwakes breeding in Svalbard migrate to the west Atlantic for the winter, which they spend at sea [34]. High levels of activity during the non-breeding season have been shown to negatively affect subsequent breeding performance in a number of seabird species [11,12,14]. We quantified kittiwakes' activity during the non-breeding season and linked this to spring migration phenology (date of arrival back to the colony), breeding phenology (lay date) and breeding performance (offspring survival) in order to measure carry-over effects. We then tested for interactions between personality and carry-over effects, to test the prediction that carry-over effects reflect differences in pace-of-life. We predicted that non-breeding activity will have negative carry-over effects on the subsequent breeding season, such that high activity will be associated with later phenology and reduced breeding performance, and that these negative carry-over effects will be stronger in shy than in bold individuals. As kittiwakes are sexually monomorphic and exhibit biparental care [35], we did not expect strong differences between the sexes, but expected that in line with other studies, carry-over effects on the timing of breeding may be stronger in females owing to greater control over egg laying.

## 2. Materials and methods

### (a) Study system

Black-legged kittiwakes lay 1–3 eggs and exhibit biparental care throughout the breeding season. We studied kittiwakes nesting on an empty building in the abandoned mining town of Grumant-byen (78°10′ N 15°05′ E), in Isfjorden on the west coast of Svalbard. Kittiwakes have been ringed and monitored during the breeding season at this site since 2008. Approximately 40 pairs breed at Grumantbyen each year. Nests were monitored from laying in early June to late chick rearing in late July. Early in the season, nests were checked weekly using a mirror mounted on the end of a pole to record the number of eggs until probable hatching time began, at which point nests were checked every 2–3 days to record the number and presence of eggs and chicks. Molecular sexing of breeding kittiwakes was conducted on DNA extracted from blood and feather samples (see the electronic supplementary material, Appendix A).

### (b) Boldness

In 2017 and 2018, we measured boldness of adult breeding kittiwakes using a novel object test, following an existing protocol [36]. Briefly, we measured individuals' response to a blue plastic penguin toy presented at the nest for 60 s, recording the proportion of the test an individual spent in each of five mutually exclusive behavioural states: (i) sitting on the nest; (ii) body raised off the nest cup, but not standing; (iii) standing on the nest (legs visible and extending to the base of the nest); (iv) off the nest but remaining on the cliff or window ledge close to the nest; (v) off the cliff or window ledge (and no longer visible). Over 2 years, 80 individuals were tested: 36 individuals were tested once, 20 were tested twice, 15 were tested three times, and nine were tested more than three times. Twenty-seven individuals were tested in both 2017 and in 2018. Using a principal component analysis (PCA), we collapsed the five behavioural variables into a single test score (PC1). This score has been shown to be highly repeatable in kittiwakes within a single breeding season ($R = 0.68$; confidence interval (CI): 0.57–0.79; $p < 0.001$; [36]). We measured adjusted repeatability (repeatability after controlling for confounding effects [37]) of PC1 across two breeding seasons using the R package *rptR* [38], including fixed effects to

Proc. R. Soc. B **287**: 20202381

adjust for a test date, breeding stage (incubation or chick rearing) and test number (the number of times an individual had previously been tested). Finally, following [39,40], we fitted a linear model with PC1 as the response variable, and individual identity (ID), test date, breeding stage and test number as fixed effects. From this linear model, we extracted parameter estimates (using the *coef*() function) for each level of the individual ID fixed effect and used these as a single estimate of boldness per individual. Parameter estimates are regarded as better estimates of individual behaviour than individual point estimates from random effects in mixed models [41]. We found no difference in boldness between the sexes (results from a linear model testing for a sex effect on boldness: $p = 0.19$).

## (c) Non-breeding activity

Between June 2012 and August 2018, adult kittiwakes were equipped with geolocator-immersion loggers of either the MK4083 series (Biotrack, $17 \times 10 \times 6.5$ mm, 1.9 g) or C65 series (Migratetech, $14 \times 8 \times 6$ mm, 1.0 g), attached to plastic leg rings. The loggers record patterns of immersion in saltwater, enabling inference of behavioural patterns in marine species. Immersion loggers were deployed with the aim of retrieval after one year to obtain data on the non-breeding period, but in some cases were retrieved after more than one year where birds were not captured during a given season (loggers retrieved after one year: $n = 71$; loggers retrieved after two years: $n = 4$). After logger retrieval, most individuals were re-equipped with a new logger to record activity during the following non-breeding season. MK4083 loggers tested for saltwater immersion every 3 s, and C65 every 30 s, both storing the sum of 'wet' readings within a 10 min bout. To facilitate comparison between logger types, we divided the values derived from MK4083 loggers by 10 such that data from both logger types ranged from 0 (continuously dry for 10 min) to 20 (continuously wet for 10 min).

Kittiwakes rest on the sea surface during the winter months, and only spend significant time on land during the breeding season, when attending their nests [34,42]. Kittiwakes are surface feeders, foraging from the surface of the water or by shallow dives from the air [35,43]. As per [43], we defined 10 min periods spent entirely dry as bouts of flight, and 10 min periods with at least 95% wet readings as bouts of resting on the sea. Ten-minute periods with 5–95% wet readings were defined as bouts of probable foraging behaviour, except in cases where a single 10 min period of intermittent wet readings occurred in between a period of flight and rest, as these are likely to indicate a period during which birds transition between flying and resting behaviours [43]. Loggers could miss bouts of behaviour shorter than 30 s in duration, but we expect such rapid shifts between flying, resting and foraging to be infrequent given the high energetic costs of taking off and landing from water in birds [44,45]. We identified the start and end of the non-breeding period for each bird using the percentage of daily time spent resting on the sea. The first day of the year on which a bird spent no time resting on the sea was regarded as its first day spent at the colony (colony arrival date), and the last day with no time spent resting on the sea as its last day at the colony (colony departure date). Each individual's non-breeding season was then defined as the interval between colony departure and arrival dates. We then extracted the daily proportion of time spent foraging, in flight, and resting, for each day of the non-breeding season. Time spent in flight and time spent resting were strongly negatively correlated ($R = -0.88$, $p < 0.001$), while there was a weaker negative correlation between time in flight and time spent foraging ($R = -0.22$, $p < 0.001$). As indicators of non-breeding activity, we averaged the daily proportion of time spent (i) foraging and (ii) in flight across all days of the non-breeding season. We interpret both time spent foraging and

time in flight as energetically costly, because we expect minimizing the time taken to acquire daily food requirements to be optimal [14]. We recorded non-breeding activity data over 78 bird-years in total, for 39 boldness-tested individuals over six years of study (22 males in 41 bird-years and 17 females in 37 bird-years), with a mean of two bird-years per individual (range 1–5 years).

## (d) Statistical analysis

Analyses were conducted in R v. 3.5.1 [46] using the *lme4* package [47] for fitting linear mixed-effects models (LMMs.) Prior to testing for carry-over effects on breeding, we first determined whether kittiwakes varied in their non-breeding activity with boldness and sex. We fitted time spent in flight and time spent foraging as response variables in two separate LMMs, with boldness, sex and their two-way interaction fitted as fixed effects, and bird ID and year fitted as crossed random effects. Because boldness tests were conducted exclusively during the final two years of tracking (2017 and 2018), there was an interval of 0–5 years between the collection of non-breeding data and boldness data (mean interval: 1.79). We therefore controlled for the interval between non-breeding period and an individual's first boldness test in these models, and found no support for an effect of interval or the interaction between interval and boldness on non-breeding activity (electronic supplementary material, Appendix B). Model selection was conducted using an information-theoretic approach, using Akaike's information criterion corrected for small sample sizes ($AIC_c$). We built a set of models from all possible combinations of predictors and refined these to a top model set by ranking according to $AIC_c$, selecting the model structure that minimized $AIC_c$ as the best model, and those within two $AIC_c$ units as competitive [48]. Because AIC can favour overly complex models [48], inference can be improved by eliminating models from the top model set if they are more complex versions of simpler (nested) models with lower $AIC_c$ values, known as the 'nesting rule' [49,50]. We therefore applied the nesting rule to prevent the retention of overly complex models, such that when two nested models differed by less than two AIC units ([51] indicating that the additional predictor has a very low explanatory power), the simplest model was preferred. When multiple models remained in the top set after applying the nesting rule, we made inference of the importance of predictors based on model-averaged parameter estimates [49].

To examine how non-breeding activity may carry-over to influence subsequent breeding, we considered effects on colony arrival date, lay date and breeding performance. Colony arrival date (days since 1 January of that year) was defined as the first day a bird spent back at the breeding colony, as identified by immersion loggers (see above). Lay date (days since 1 January of that year) was defined as the first day on which a bird's nest contained an egg. Breeding performance was represented by the number of days survived by birds' offspring. In separate LMMs with Gaussian distributions, we fitted colony arrival date, lay date and offspring survival as response variables and included the following predictors: (i) time in flight, (ii) time foraging, (iii) boldness and the two-way interactions between (iv) time in flight and boldness and (v) time foraging and boldness. Because the date of arrival to the breeding colony can influence the timing of breeding, and both the timing of arrival and of breeding can influence breeding success [8], we additionally included (vi) colony arrival date as a fixed effect in lay date and offspring survival models, and (vii) lay date in offspring survival models. Colony arrival date and lay date were weakly correlated (electronic supplementary material, Appendix C), but for all models, we inspected variance inflation factors (VIFs) of predictor variables and found no evidence of collinearity (VIFs < 2.5 in all cases, indicating minimal collinearity [52]).

Boldness was fitted as a continuous measure in all analyses and was grouped in figures for illustrative purposes only. Bird ID and year were fitted as crossed random effects. We ran all carry-over effects models separately for males and females to control for non-independence of breeding outcomes between paired birds. Model selection was conducted using $AIC_c$, as specified above. We calculated the marginal coefficient of determination ($R^2_m$, variance explained by fixed effects) and the conditional coefficient of determination ($R^2_c$, variance explained by both fixed and random effects) for all top-ranking models using the *MuMIn* package [53] (electronic supplementary material, Appendix D). Full model tables are presented in the electronic supplementary material, Appendix E. All variables were standardized (to a mean of 0 and standard deviation of 1) to facilitate model fitting and interpretation of results. Boldness was reflected and square-root transformed to adjust for negative skewness, and then reflected back to the original direction, to meet normality assumptions. Additionally, in order to further explore whether our results were affected by the interval between the collection of non-breeding data and boldness data, we reran all carry-over effects models on a subset of the data where this interval was two years or less. Parameter estimates from the conservative data subset were similar to those from the full dataset in both strength and direction, and are presented in Appendix F of the electronic supplementary material.

## 3. Results

### (a) Boldness

PC1 explained 58.37% of the variation in response to the novel object, and across two years individuals were highly repeatable in their test responses ($R = 0.61$, CI: 0.48–0.73; $p < 0.001$). Boldness scores were inverted such that low values represented when birds left the nest (interpreted as 'shy' responses), and high values represented when birds remained sitting on the nest (interpreted as 'bold' responses; table 1). Boldness scores ranged from −0.86 to 1.36. These results are comparable with findings from a single year of personality testing on black-legged kittiwakes [36].

### (b) Variation in non-breeding season activity

We did not find an effect of boldness, sex, or their two-way interaction on kittiwake non-breeding activity: the best-supported models predicting variation in both time spent foraging and time in flight during the non-breeding season contained only model intercepts.

### (c) Boldness and carry-over effects on breeding

Arrival date back to the colony in spring was related to the interactions between boldness and winter activity for male kittiwakes, but not for females (table 2). Among males, steeper slopes between arrival date and time spent foraging and in flight (figure 1a,b) indicate negative carry-over effects were strongest in shyer individuals. Among female kittiwakes, time spent in flight also predicted later return to the colony (table 2 and figure 1c), but this negative carry-over did not interact with boldness.

For lay date models, winters characterized by more time foraging and in flight were followed by later started clutches among males (table 2 and figure 1e,f), but the interaction with boldness was not supported. Meanwhile for females, more time spent in flight interacted with boldness, predicting

**Table 1.** Variable loadings and cumulative variance explained for each principal component of the boldness test principal component analysis.

| behaviour | PC1 | PC2 | PC3 | PC4 | PC5 |
|---|---|---|---|---|---|
| sitting | −0.77 | −0.38 | 0.17 | 0.16 | −0.45 |
| raised up | 0.06 | 0.71 | 0.51 | 0.18 | −0.45 |
| standing | 0.06 | 0.19 | −0.80 | 0.38 | −0.45 |
| off the nest | 0.03 | 0.04 | −0.13 | −0.88 | −0.45 |
| off the ledge | 0.63 | −0.56 | 0.24 | 0.19 | −0.45 |
| cumulative variance explained | 0.58 | 0.85 | 0.96 | 1.00 | 1.00 |

later egg laying most strongly in shy individuals (table 2 and figure 1g).

A large number of variables were supported in offspring survival models, but they explained little variation in the data (table 2). For males, offspring survival was related to the interaction between boldness and time spent foraging in winter, with stronger negative effects for shy than bold individuals (table 2 and figure 1j). More time spent in flight was associated with higher offspring survival for males, while for females, offspring survival was positively related to time spent foraging, but negatively associated with time in flight (table 2; figure 1i–l).

Coefficients of determination indicated that the variation explained by fixed effects was between 8 and 22% for colony arrival date, 8–36% for lay date and 1–12% for offspring survival (see the electronic supplementary material, table D1). This suggests that most of the variation in breeding was explained by differences among individuals and years, particularly in offspring survival models.

## 4. Discussion

Carry-over effects link individuals' activity during one season to their performance in subsequent seasons, but despite being measured at the individual level, the examination of the factors shaping individual differences in carry-over effects has been lacking. This study is, to our knowledge, the first to investigate how carry-over effects are influenced by personality, and to demonstrate personality-specific carry-over effects. We find sex- and personality-dependent carry-over effects of non-breeding activity in kittiwakes. Males that spent more time foraging during the non-breeding season arrived back later to the colony the following spring, began breeding later and had lower offspring survival. For female kittiwakes, more time spent in flight was associated with later colony arrival, later egg laying and lower offspring survival, while time spent foraging had a positive effect on offspring survival. Interactions between boldness and non-breeding activity supported personality-dependent carry-over effects, and in all supported interactions, we found that negative carry-over effects were stronger in shy individuals than in bolder individuals. These results are in line with predictions that personality should be linked to life-history trade-offs and emphasize the importance of considering interactions with intrinsic factors when determining the consequences of carry-over effects for population dynamics.

**Table 2.** Model-averaged estimates from the best-supported models investigating the effects of winter activity and boldness on the subsequent breeding season. (Best-supported models were those retained where $\Delta$AICc < 2 and where there was no simpler outranking model (the 'nesting rule', [49]). Model-averaged estimates ± standard errors are reported for predictors retained in best-supported models only. Importance is the relative variable importance, calculated as the sum of Akaike weights of the models in which that term appears. Bird ID and season were fitted as crossed random effects in all models. Arrival date and lay date were controlled for in offspring survival models, and arrival date was controlled for in lay date models (these variables are in grey for their own respective models where they were not fitted as fixed effects). See the electronic supplementary materials table D1 for summaries of best-supported models including coefficients of variation, and tables E1–E3 for full model outputs.)

| | predictor | colony arrival date est ± s.e. | importance | lay date est ± s.e. | importance | offspring survival est ± s.e. | importance |
|---|---|---|---|---|---|---|---|
| males | intercept | 118.41 ± 2.30 | — | 162.94 ± 1.33 | — | 13.02 ± 2.95 | — |
| | boldness | 0.28 ± 0.99 | 0.52 | | 0.00 | −0.36 ± 1.23 | 0.42 |
| | foraging | 2.12 ± 0.95 | 1.00 | 1.97 ± 0.95 | 1.00 | −2.06 ± 1.32 | 0.90 |
| | flight | 0.00 ± 1.04 | 0.52 | 1.40 ± 0.95 | 1.00 | 1.39 ± 1.44 | 0.36 |
| | boldness × foraging | −2.15 ± 1.03 | 0.52 | | 0.00 | 2.13 ± 1.32 | 0.42 |
| | boldness × flight | −2.06 ± 1.10 | 0.52 | | 0.00 | | 0.00 |
| | arrival date | | | | 0.00 | | 0.00 |
| | lay date | | | | | −1.31 ± 1.41 | 0.58 |
| females | intercept | 119.16 ± 1.33 | — | 161.63 ± 1.33 | — | 14.52 ± 3.62 | — |
| | boldness | 1.97 ± 0.94 | 1.00 | 2.76 ± 0.63 | 1.00 | −1.75 ± 1.37 | 0.59 |
| | foraging | | 0.00 | | 0.00 | 1.50 ± 1.32 | 0.41 |
| | flight | 2.48 ± 0.88 | 1.00 | 2.96 ± 0.79 | 1.00 | −1.02 ± 1.50 | 0.14 |
| | boldness × foraging | | 0.00 | | 0.00 | | 0.00 |
| | boldness × flight | | 0.00 | −1.77 ± 0.62 | 1.00 | | 0.00 |
| | arrival date | | 0.00 | | | −1.10 ± 1.52 | 0.07 |
| | lay date | | | | | −1.35 ± 1.50 | 0.37 |

## (a) Carry-over effects of non-breeding activity

There is increasing evidence that activity during the non-breeding season influences subsequent breeding performance [11,12,14,54], facilitated by advances in biologging technology. In concordance with a number of other studies on seabirds [11,12,14], we detected predominantly negative carry-over effects of time spent both flying and foraging on subsequent breeding performance. Among males, spending more time foraging during the winter preceded later arrival back to the colony, later started clutches and lower offspring survival; more time spent in flight was also associated with later laid eggs and lower offspring survival in males. Among females too, winters characterized by more time in flight preceded later return to the colony, later egg laying and lower offspring survival. This suggests, in accordance with previous work [11,12,14], that seabirds increase activity during the winter to compensate for poor foraging conditions, or for their own poor body condition [12]. Individuals in poor condition may be forced to prolong their time at wintering grounds in order to attain condition sufficient for breeding [55], resulting in later return to the breeding grounds and later onset of breeding, and, if sufficient condition is not reached, reduced breeding success [56,57]. Further, numerous studies have linked both poor winter body condition to reduced probability of attempting to breed at all the following season [10,56,58]. However, we were unable to test whether non-breeding activity influenced the probability of skipped breeding, owing to a lack of data on individuals that did not attempt breeding in a given year. While we detected exclusively negative carry-over effects of non-breeding activity on breeding phenology, higher offspring survival was predicted by more time spent in flight in males, and more time spent foraging in females. One potential explanation for where non-breeding activity negatively impacted phenology yet positively affected offspring survival is that increased effort can successfully compensate for poor conditions enough to improve chick-rearing performance, even if poor conditions results in delayed arrival.

## (b) Personality-dependent carry-over effects

Time spent in flight and time spent foraging during the winter were both unrelated to individuals' boldness scores. All observed negative carry-over effects were stronger in shyer individuals than in bolder birds. Winters characterized by high activity were followed by later return to the colony and lower offspring survival in shy males, and later egg laying in shy females, but these effects were attenuated in bolder individuals, suggesting that variation in boldness is associated with differential breeding responses to non-breeding conditions. The directionality of these findings is consistent with the pace-of-life syndrome hypothesis, which predicts a coupling between life-history and personality, such that a fast pace-of-life should be associated with boldness and a slow pace-of-life with shyness [21]. Under challenging conditions, the trade-off between allocation to self-maintenance and to the current reproductive effort is exacerbated [59], forcing

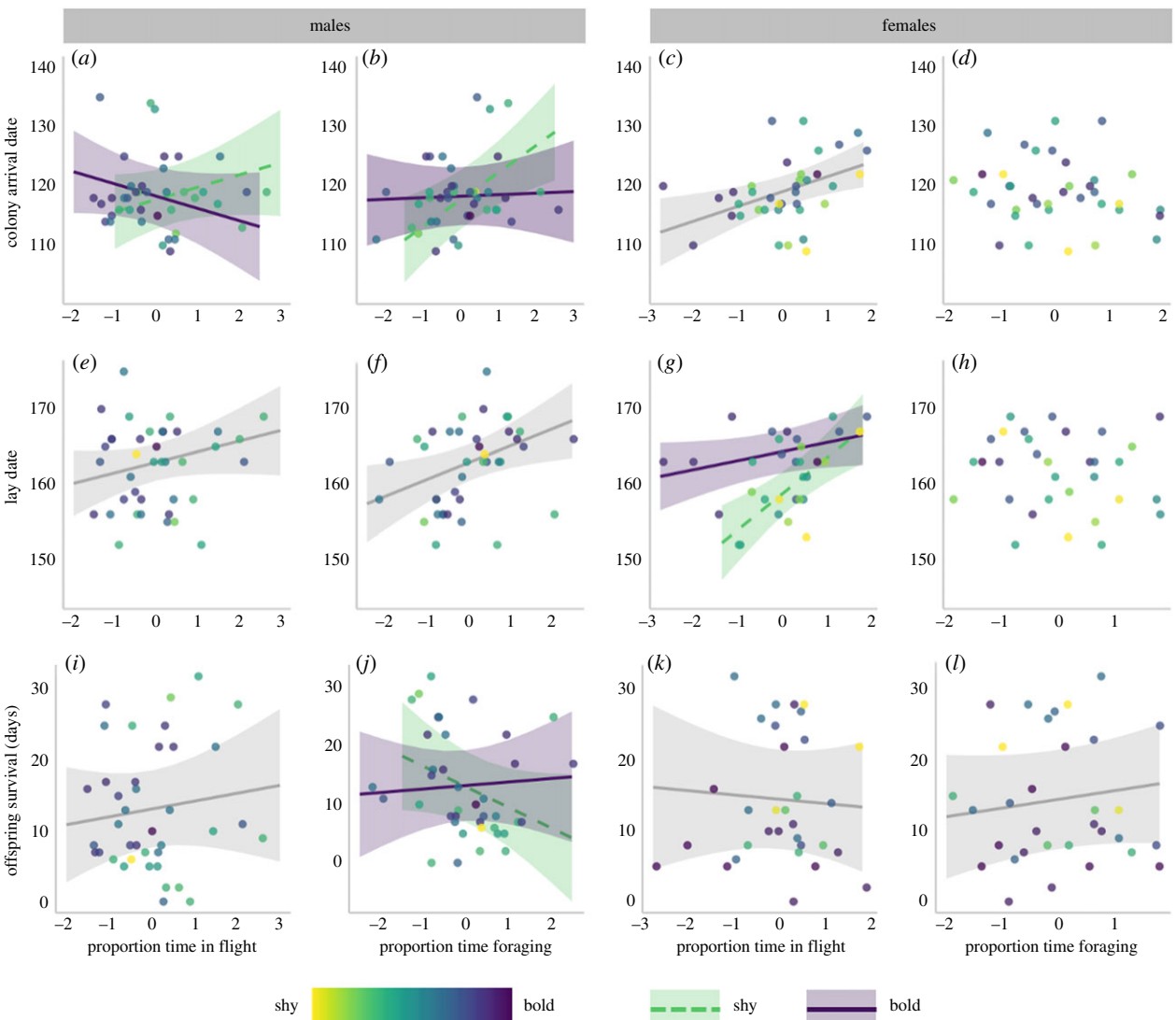

**Figure 1.** Carry-over effects of non-breeding activity (time spent in flight and time spent foraging) for male (left two columns) and female (right two columns) kittiwakes. Top row: carry-over effects on colony arrival date (days since 1 January); middle row: carry-over effects on lay date (days since 1 January); bottom row: carry-over effects on offspring survival (number of days since hatching). Point colour represents boldness from boldest (purple) to shyest (green). Boldness is fitted as a continuous measure in all analyses. For plotting purposes only, where an interaction between boldness and activity was supported, estimates are presented for the boldest individuals (+1 standard deviation from the mean) in purple solid lines, and for the shyest individuals (−1 standard deviation from the mean) in green dashed lines. A single line indicates no interaction between activity and boldness, and no line indicates no effect of activity on arrival date. Shaded area represents 95% confidence intervals. (Online version in colour.)

individuals to make decisions between allocating to one over the other. Our findings suggest that shy individuals may be more likely to respond to poor condition by allocating away from reproductive activities and instead towards self-maintenance. This may be achieved by spending longer at the wintering grounds [55], in order to spend more time foraging to regain lost condition, with detrimental effects on the timing of breeding and on breeding performance. In more extreme cases, where conditions are particularly poor, shy individuals may also be more likely to skip breeding for a year altogether. Bold individuals' breeding performance and phenology was less dependent upon non-breeding activity, suggesting that bold individuals' breeding strategies involve high allocation to breeding attempts, irrespective of costs to an individual's condition. Interestingly, following what we interpret as 'good' non-breeding conditions (when birds spent less time foraging and in flight), shy individuals performed equal to or even better than bold individuals. For example, shy males arrived earlier to the colony and had higher offspring survival following winters when they spent

less time foraging and in flight. This suggests that bold and shy birds did not differ in quality, but in how they respond to non-breeding conditions.

The pace-of-life syndrome hypothesis has mixed support, with a recent meta-analysis demonstrating that evidence for correlations between individual behaviour and life-history is weak, particularly in vertebrate species [60]. However, a recent review highlighted that a lack of support for the pace-of-life syndrome hypothesis may be owing to phenotypic plasticity in response to the environment obscuring a clear link between personality traits and reproductive output [61]. Testing for a relationship between personality and breeding performance contingent on an individual's condition may remove confounding effects of environmental variation on breeding. A strong relationship between boldness and lifetime reproductive success, especially in species with restricted breeding opportunities, would probably lead to strong, directional selection and elimination of variation in boldness. Alternatively, an effect of personality on condition-dependent reproductive

performance, as reported here, may result from behavioural life-history syndromes and evade directional selection.

Carry-over effects have also been found to vary with age in some species [11]. In wandering albatrosses (*Diomedea exulans*), while younger birds all bred successfully, high foraging effort during the winter months was linked to increased breeding failure in older individuals [11]. Age differences may thus also explain variation in carry-over effects in our study, although we were unable to explore this possibility because birds in this population were of unknown age. While personality traits have been found to be stable over long periods (e.g. [62]), directional changes in boldness have also been documented in some species. Thus, the relationship between boldness and carry-over effects in kittiwakes could be linked to age differences if kittiwakes become shyer in older age. Theory and empirical findings generally suggest the opposite pattern, whereby animals get bolder with age, as their residual reproductive value decreases [62–64]. Nevertheless, further research should investigate the relationship between age, boldness and the strength of carry-over effects on breeding.

Carry-over effects may also interact with boldness by acting upon personality traits directly. Personality traits are typically characterized by their stability, but recent work has recognized the importance of within-individual changes in personality in response to environmental conditions, known as behavioural plasticity [65,66]. Our method of assaying boldness captures individuals' propensity to defend their nest, and we may therefore expect that when carry-over effects of winter conditions lead an individual to invest less in reproductive performance, they should also behave more shyly. By assaying boldness in individuals over periods of several more years, it would be possible to quantify individuals' plasticity in personality in relation to non-breeding conditions and test whether carry-over effects also act upon personality traits. Furthermore, using longitudinal boldness data, future work could test whether individuals consistently differ in their plasticity in response to winter conditions [65] and examine whether plasticity in personality is adaptive, and its consequences for lifetime fitness.

## (c) Sex-specific carry-over effects

A number of studies in birds have reported that carry-over effects on the timing of breeding are stronger in females than in males [26–29], attributing this to female control over the timing of egg laying [67]. Here, we found that in kittiwakes, the timing of laying was related to the non-breeding activity of both sexes. This implies that the timing of laying is driven by both female and male condition: males in better body condition may advance their partner's lay date through earlier engagement in breeding behaviours such as nest building, courtship feeding and, ultimately, copulation [68]. However, only among females did we detect an interaction between boldness and non-breeding activity on lay date. While male condition may influence the timing of breeding activities, the interaction between boldness and non-breeding activity among females may suggest that females are better able to optimize the timing of laying to their pace-of-life.

Male and female kittiwakes showed differences in the non-breeding behaviours that influenced their subsequent breeding phenology and performance. For males, the strongest carry-over effects were of time spent foraging, while among females, time spent in flight affected phenology, but foraging did not. Furthermore, more time spent in flight preceded later breeding, but higher offspring survival in males, while in females, more time spent foraging improved offspring survival. This sex difference in the non-breeding behaviours driving carry-over effects may be the result of a number of behavioural and physiological inequalities between males and females [27,28]. First, kittiwakes may exhibit sex-dependent non-breeding foraging strategies. Focusing solely on the carry-over effects on offspring survival suggests that spending more time in flight and less time foraging is beneficial to males, while in females, we observed the opposite effect, with spending more time foraging and less time in flight apparently optimal. This pattern could suggest trade-offs between the ability to successfully locate and obtain food, with successful males being less efficient at finding prey but more efficient at capturing it, and the reverse being true for successful females. Second, owing to sex-specific breeding roles, males and females may differ in their energetic requirements for breeding. Male kittiwakes may be more limited by winter foraging activity if their energetic requirements are higher than that of females, for example, owing to their slightly larger body size [35]. Other studies on sexually monomorphic seabirds have also reported unexpected sex-specific carry-over effects [69], and closer examination into the year-round activities of such species is required to elucidate the mechanism driving these relationships. Regardless of their cause, sex-dependence adds an additional layer of complexity to carry-over effects, with consequences for sexual selection, and for population-level dynamics [70].

**Ethics.** All bird handling, sampling and logger deployment was carried out with approval and permits granted by the Norwegian Food Safety Authority (permits 8602 and 8616) and the Governor of Svalbard (Norway).

**Data accessibility.** Data are accessible from the Dryad Digital Repository: https://dx.doi.org/10.5061/dryad.g79cnp5nq [71].

**Authors' contributions.** S.M.H., S.D., L.U.S. and S.C.P. conceived the study. S.M.H., S.D., P.B. and S.C.P. collected the data. M.C. wrote the initial script for the analysis of activity data from saltwater immersion loggers. S.M.H. analysed the data and wrote the first draft. All authors contributed to the preparation of the manuscript.

**Competing interests.** We declare we have no competing interests.

**Funding.** This study was funded by MOSJ, SEAPOP and NERC (grant no. NE/L002450/1).

**Acknowledgements.** We thank Tommy Clay, Francoise Amelineau and Benjamin Merkel for advice on analysis of saltwater immersion data. We are grateful to Tommy Clay, Jon Green, Ruth Dunn, Jamie Duckworth, Teri Jones, Finn McCully, Lila Buckingham, Sophie Bennet and Kit Maskrey for useful conversations and providing feedback on this manuscript. We thank the many fieldworkers who supported data collection in Svalbard, and the Norwegian Polar Institute for logistical field support. Thanks to Oddmund Kleven (Norwegian Institute for Nature Research; NINA) for molecular sexing of birds.

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
