## [Reviewer comments · Proceedings of the Royal Society B: Biological Sciences]

Review History

RSPB-2020-0324.R0 (Original submission)

Review form: Reviewer 1

Recommendation

Major revision is needed (please make suggestions in comments)

Scientific importance: Is the manuscript an original and important contribution to its field?

Good

General interest: Is the paper of sufficient general interest?

Excellent

Quality of the paper: Is the overall quality of the paper suitable?

Good

Is the length of the paper justified?

Yes

Should the paper be seen by a specialist statistical reviewer?

Yes

Do you have any concerns about statistical analyses in this paper? If so, please specify them explicitly in your report.

Yes

It is a condition of publication that authors make their supporting data, code and materials available - either as supplementary material or hosted in an external repository. Please rate, if applicable, the supporting data on the following criteria.

Is it accessible?

N/A

Is it clear?

N/A

Is it adequate?

N/A

Do you have any ethical concerns with this paper?

No

Comments to the Author

This study tested for sex- and personality-specific carry-over effects from winter to the following breeding season in black-legged kittiwakes. It found, as predicted, that carry-over effects (timing of breeding and breeding success) were stronger in shy individuals than in bold individuals. It also found opposite effects in males and females.

The paper is well written, except for a few typos. I could not evaluate the use of the literature because the citations in the text are numbered but the reference list is in alphabetical order without numbers.

I have a few questions and concerns about the methods and interpretation of the results:

Lines 124-125: Adjusted repeatability how and why?

Lines 127-128: I don't follow this. What exactly was used from the regression as the measure of boldness? Why is individual ID a fixed effect and not a random effect? Does this model allow for individual slopes? How does it account for differences in the number of times individuals were tested?

Lines 139-141: This could be biased if short flights or immersions were missed by the C65 tags. Does it change anything if the data from the MK4083 tags are subsampled at 30-second intervals? If 30 seconds is shorter than any activity bout, state that.

Lines 164-165: Are boldness and sex correlated? You state that they are not (at least not enough to affect the statistics) at the end of this section, but it would help to state it here, so the reader isn't distracted by wondering about it while reading the rest of the methods.

Line 171: I'm not familiar with this nesting rule. It seems that you don't have that many predictors and they are potentially important ones. How would results change if you didn't apply the nesting rule?

Line 181: Time in flight and time foraging should be negatively correlated, as they are mutually exclusive activities. Only if they are each small proportions of total time would they not be correlated. Please expand on this.

Line 181: Again, please tell us here that predictors were not correlated with each other instead of waiting until the end of the section.

Lines 184-185: I find it surprising that colony arrival date and laying date are not significantly correlated for females, regardless of what the variance inflation factor is. I would like to see a graph of laying date plotted against arrival date. This might be a supplemental figure.

Lines 192-194: Put this earlier (see above).

Lines 206-208: I don't understand the nesting rule well enough to evaluate this result. Without the nesting rule, models with boldness only and with sex only are competitive with the "best" model

for time in flight. Ignoring those models has possible implications for the carry-over effects models.

Lines 211-212: It is not clear to me that this is the correct interpretation. For males, a competing model for arrival date did not include boldness. For both sexes, several competing models for breeding success did not include boldness.

Lines 254-255: Please explain why you think carry-over effects of time flying and foraging were similar. If I understood correctly, effects of foraging activity differed between the sexes and the two activities had opposite effects on females.

General comments:

Line 81: In most seabird species, males and females both incubate eggs and rear young, so at least behaviorally, there is little difference in breeding roles. Relative energetic investment in breeding between the sexes is debated.

Lines 298-300: The correct metric here would be lifetime reproductive success. If shy individuals have lower average success than bold individuals, but breed longer, lifetime reproductive success could be equal.

Tables and Figures:

Table 2: It would be helpful to show all models with $\Delta AICc$ within 2 (those models that were dropped under the nesting rule).

Tables 2 and 3: Please explain what the gray areas mean. The reader can figure it out, but it would be simpler if you explained it.

All figures: I cannot tell the shades of dots apart. Please use more color contrast and larger dots so the colors show more.

Editorial comments:

Line 22: Delete "of" from "effects of on breeding".

Line 56 & others: Spell out POLS. We don't need more acronyms that just make it harder for nonspecialists to read the paper.

Line 98: "great control" should perhaps be "greater control" as it is in relationship to males' control.

Lines 155-158: The second sentence seems to mostly repeat the first sentence. Either combine the two sentences or reword to make it clear how they differ.

Lines 212-213: Figure 3 is called out before Figure 2. They should be reversed so Figure 2 is called out first.

Line 242: "high flying activity" is a kind of ambiguous term (sounds like flight altitude). Perhaps "spending more time flying" would be clearer.

Line 246: Please spell out pace-of-life syndrome. If you don't spell it out in the rest of the discussion, please do so here, because I already forgot what POLS stands for by the time I got to the discussion.

Lines 307-308: Please delete extra words in this sentence.

Review form: Reviewer 2

Recommendation

Major revision is needed (please make suggestions in comments)

Scientific importance: Is the manuscript an original and important contribution to its field?

Excellent

General interest: Is the paper of sufficient general interest?

Excellent

Quality of the paper: Is the overall quality of the paper suitable?

Good

Is the length of the paper justified?

Yes

Should the paper be seen by a specialist statistical reviewer?

No

Do you have any concerns about statistical analyses in this paper? If so, please specify them explicitly in your report.

No

It is a condition of publication that authors make their supporting data, code and materials available - either as supplementary material or hosted in an external repository. Please rate, if applicable, the supporting data on the following criteria.

Is it accessible?

N/A

Is it clear?

N/A

Is it adequate?

N/A

Do you have any ethical concerns with this paper?

No

Comments to the Author

The major goal of this research is to test whether individual differences in personality traits (particularly in bold-shy axis) influence the strength of carry-over effects on breeding performance. More specifically, the authors test a hypothesis deriving from pace-of-life syndrome: carry-over effects on breeding performance are weaker in bold (fast-paced) individuals, as they tend to maintain an allocation to reproduction irrespective of previous conditions, while shy (slow-paced) individuals experience stronger carry-over effects because they favor allocation to self-maintenance over current reproduction. They tested this hypothesis in black-legged kittiwakes, which exhibit biparental care behavior. First of all, they discovered that a negative carry-over effect of non-breeding activity on breeding performance. Second, they further revealed that carry-over effects of non-breeding activity on breeding performance were stronger in shyer individuals: active winters were followed by later breeding onset and worse breeding performance in shy birds, but these effects were weaker or undetected in bolder individuals. Overall, this is a great study and I enjoy reading this manuscript, but there are a few places need to be addressed or further clarified in the article. Following are some major concerns and minor suggestions to the authors:

1. Boldness traits in statistical analysis:

According to the authors' description in the manuscript, they assessed boldness traits using a novel object test in 2017 and 2018, and found these scores are highly repeatable within a single breeding season. Here are some of my concerns:

(1) did the authors only measure boldness in 2017 and 2018 and use this boldness score to fit all the linear mixed-effects models? As far as I read, the non-breeding activity (perhaps also breeding performance) was collected between 2012 and 2018, which means the authors used the boldness data from the later time point to predict the influences on breeding performance in earlier times (2012-2016). My question is that how repeatable this boldness index across different

breeding seasons? A challenge when studying pace-of-life syndrome is the fact that behavior traits or physiological traits are highly labile, which means they are sensitive to study methods and even to environmental variation (Beckmann and Biro, 2013; Carter et al., 2013; Biro, Adriaenssens and Sampson, 2014). I understand that personality traits like boldness are considered as stable traits, but they still can vary across at different time points. Use the single-year boldness score (e.g., boldness score in 2017-2018) as an independent variable to test its influence on breeding success across several years (e.g., breeding performance in 2012-2016) might not be an ideal setup.

References mentioned here:

Beckmann C and Biro PA (2013). *Ethology*, 119, 937-947.

Carter AJ, Feeney WE, Marshall HH, Cowlshaw G and Heinsohn R (2013). *Biological Reviews*, 88, 465-475.

Biro PA, Adriaenssens B and Sampson P (2014). *Journal of Animal Ecology*, 83, 1186-1195.

(2) The authors mentioned there are 80 individuals were tested in boldness test. Are these the same group of animals sampled for non-breeding activity and scored breeding performance? If yes, how many males and females in this population?

(3) Are all these 5 behaviors representing boldness? Which are the loading weights of each behavior in the PC1? I usually would recommend including a table for principal component analysis output in supplementary data to illustrate the cumulative variance explained in all PCs, just like what the authors did in their earlier publication in 2020 (Harris et al. (2020) *Animal Ecology* 89:68-79).

2. Data interpretation:

The authors indicated that their results revealed predominantly negative effects of winter activity on breeding performance, which I totally agree with it. However, there are some positive carry-over effects on breeding. For example, in males, "flight" activity in the non-breeding season has a positive effect on offspring survival; in females, "forage" activity in non-breeding season also has a positive effect on offspring survival. I think the authors did not further explain these findings. I also think the authors should discuss why these two non-breeding activity traits (forage and flight) sometimes have opposite effects on the same breeding traits. To me, one thing that can be discussed is that, although the authors treated both "forage" and "flight" as the non-breeding activity, they may affect breeding in different ways because the "flight" is an energy-consuming behavior, while "forage" is an energy-refilled behavior. I think that is why we sometimes see the opposite effects of these two behaviors on breeding performance. For instance, in the offspring survival model, forage behavior is positively associated with offspring survival, but flight behavior is negatively associated with offspring survival, though we see both behaviors are negatively correlated with colony arrival date and lay date.

3. Materials & Methods:

The author did not describe how do they access breeding performance and breeding success in this section. There is no clear definition for some important terms, like "colony arrival date" and "lay date". Is the date starting from the first day of each year? Or it is starting from the last day of the previous breeding season? Also, what are the relationships of these two traits (colony arrival date and lay date) and breeding performance? I am a bit confused at the beginning because these two traits are negatively correlated with breeding success, but offspring survival is positively correlated with breeding success.

4. Results and Tables:

I like the way the authors used Akaike's Information Criterion (AICc) to test and select LMM models! But in the final model (Table 3), I think the authors should also report F-value and p-value for all fixed variables and interaction terms. In the Results section and Tables, there are no p-values reported in context. I understand p-value is not everything, but without it, it is hard to convince the readers that which variables have significant effects on responding variables. In the

figures, since the authors use the LMM regression models, it would be better to report the estimate (slope), R²-values, p-values of the best-fitted lines so that it would be easy for readers to understand how the strong/weak the relationships are.

Minor comments:

Line 66-69:

If at the population levels, previous study has shown the fast-paced and slow-paced individuals are responding differently to stress in reproductive behavior, what is the advantage of doing similar experiment at individual levels? I think the authors can further elaborate this more to emphasize the importance of their research.

Line 95:

But in the present setup, the authors cannot compare the sex differences because (1) they used different model components in males versus in females; (2) they analyze males and females separately.

Line 110-111:

The authors should also include the protocol of how they conduct blood collecting and DNA extraction for identifying sex in the Methods.

Line 159:

How many males and females in these 49 individuals? Do all these individuals have boldness data from each year? Please also see the first point in the major concerns.

Line 182-185:

It seems that "arrival date" and "lay date" are associated with one of each other. Is there any multi-collinearity issue in this model?

Line 210:

I understand that sex variable can be separately analyzed in "colony arrival" and "lay date" models, but "offspring survival" are contributed by both males and females because this species exhibit biparental care (as the authors mentioned in previously), and therefore separately analyzing sex might raise some issues since you wouldn't be able to count the interactions between the male and female.

Line 212:

Please also report statistical results (e.g., F-value and p-value) of each result to support the statement. Please also see the fourth point in the major concerns.

Line 213:

Please specify whether it is a positive or negative association.

Line 216:

But in both males and females, the authors also find positive carry-on effects on offspring survival. Please also see the second comment in the major concerns.

Line 225-231:

Do all these effects reach statically significant? I understand p-value is not everything, but if it is applicable, the authors should report the statistical results to back up their statement.

Line 244-245:

Not exactly true. In the offspring survival model, the "flight" and "forage" traits have opposite effects on offspring survival.

Line 245:
Any statistical analysis can support this statement?

Line 287-288:
Which part of data can support this statement?

Line 322-324:
Did the author have "size" data to incorporate into their analysis?

Reference:
Please add reference numbers into the reference list. Otherwise, it is difficult to track the citations which are mentioned in the context.

Line 338: "fo 67:1-48"
Is "fo" a typo?

Tables:
Did all analysis include "bird identity" and "year" as random variables? If yes, please specify in the table legend.

Table 1 and 2
The "X" and "-" is confusing. The authors can leave the non-included variable as blank. Also, I would recommend moved Table 1 and 2 to supplementary data. Moreover, in addition to estimate and importance, the authors should also report F-values and p-values (if applicable) of each variable. Please also see the fourth comment in the major concerns.

Figures:
Please also report slope of the best-fitted line, R2-values, F-values and p-values in either the figure or in the context.

Decision letter (RSPB-2020-0324.R0)

09-Apr-2020

Dear Ms Harris:

I am writing to inform you that your manuscript RSPB-2020-0324 entitled "Personality-specific carry-over effects on breeding" has, in its current form, been rejected for publication in Proceedings B.

This action has been taken on the advice of referees, who have recommended that substantial revisions are necessary. With this in mind we would be happy to consider a resubmission, provided the comments of the referees are fully addressed. However please note that this is not a provisional acceptance.

Sincerely,
 Dr Maurine Neiman
 mailto: proceedingsb@royalsociety.org

Associate Editor

Comments to Author:

This is a neat combination of large-scale field datasets, and a pleasure to read. The carry-over effects are interesting, even before the contribution of personality. I agree with the referees that areas need clarification in order to fully assess the manuscript. If methodological points of referee 1 can be addressed, then I find the links between over-winter behaviour and reproduction compelling. However, the key aspect is personality. And I share the concern raised by Referee 2 on the timing of boldness assays.

While the emphasis of the manuscript is directional (personality mediates carry-over effects), the over-winter data appear to be collected up to 6 years before the boldness tests. First, an alternative explanation is therefore carry-over effects of over-winter behaviour on both boldness and reproduction. Indeed, it would be really interesting if this were the case. While the lack of correlation in the first model suggests it's not, interval (between over-winter and boldness measurements) should be controlled for in this model. If data allow, a stronger option would be to explicitly correlate over-winter behaviour in the 2017-2018 winter to change in boldness between these years.

Second, the high over-year repeatability is remarkable – clearly a striking behavioural difference. But I am concerned as to whether this is evidence enough of consistency for the period of study, where 6 years is a large part of the average post-recruitment lifespan. For example, are age-related shifts in reproductive investment described in kittiwakes? It is possible that repeatability in boldness, which is effectively investment in nest defence, is inflated by comparing adjacent years within relatively young versus old individuals. This becomes an issue if long intervals are prevalent in the dataset – which may not be the case – as there is then a risk that relatively young (less efficient?) or middle aged (peak performance?) winter strategies are being correlated to characteristically 'middle-aged' versus 'old' personality types. Adding interval to analyses and/or re-running with a conservative subset of the data that have a short interval between measurements would be useful.

These concerns may be simply addressed by explaining the data distribution, which is not quite clear in the manuscript. Specifically, the number of birds with over-year replication in the boldness test (lines 120 and 200), the average number of over-winter measurements per bird (range 1-4 on lines 136/159) and, importantly, average/range of interval. As a ringed population, is age or size known?

It is vitally important to make use of opportunities to link existing datasets and explore longer term patterns. But then limitations that arise should be discussed clearly throughout. If it is not possible to address these concerns with the existing data, I would recommend revising the manuscript to give alternative explanations some or potentially equal weight.

Overall, I agree with the referees that this is a thought-provoking study, but further information or analyses are required to evaluate the results. I hope the referees' clear and supportive comments will be useful in revision. One to note is that both indicate uncertainty over the strength of effects - Referee 1 in terms of the clarity of figures and Referee 2 the goodness of fit. I had a similar sense, as it is not clear in Figures 1a, 1c and Figure 3 how the underlying datapoints are described by the fitted lines (particularly where they extend into areas without data). Thus, it may be necessary to consider how best to present the results.

Reviewer(s)' Comments to Author:

Referee: 1

Comments to the Author(s)

This study tested for sex- and personality-specific carry-over effects from winter to the following breeding season in black-legged kittiwakes. It found, as predicted, that carry-over effects (timing of breeding and breeding success) were stronger in shy individuals than in bold individuals. It also found opposite effects in males and females.

The paper is well written, except for a few typos. I could not evaluate the use of the literature because the citations in the text are numbered but the reference list is in alphabetical order without numbers.

I have a few questions and concerns about the methods and interpretation of the results:

Lines 124-125: Adjusted repeatability how and why?

Lines 127-128: I don't follow this. What exactly was used from the regression as the measure of boldness? Why is individual ID a fixed effect and not a random effect? Does this model allow for individual slopes? How does it account for differences in the number of times individuals were tested?

Lines 139-141: This could be biased if short flights or immersions were missed by the C65 tags. Does it change anything if the data from the MK4083 tags are subsampled at 30-second intervals? If 30 seconds is shorter than any activity bout, state that.

Lines 164-165: Are boldness and sex correlated? You state that they are not (at least not enough to affect the statistics) at the end of this section, but it would help to state it here, so the reader isn't distracted by wondering about it while reading the rest of the methods.

Line 171: I'm not familiar with this nesting rule. It seems that you don't have that many predictors and they are potentially important ones. How would results change if you didn't apply the nesting rule?

Line 181: Time in flight and time foraging should be negatively correlated, as they are mutually exclusive activities. Only if they are each small proportions of total time would they not be correlated. Please expand on this.

Line 181: Again, please tell us here that predictors were not correlated with each other instead of waiting until the end of the section.

Lines 184-185: I find it surprising that colony arrival date and laying date are not significantly correlated for females, regardless of what the variance inflation factor is. I would like to see a graph of laying date plotted against arrival date. This might be a supplemental figure.

Lines 192-194: Put this earlier (see above).

Lines 206-208: I don't understand the nesting rule well enough to evaluate this result. Without the nesting rule, models with boldness only and with sex only are competitive with the "best" model for time in flight. Ignoring those models has possible implications for the carry-over effects models.

Lines 211-212: It is not clear to me that this is the correct interpretation. For males, a competing model for arrival date did not include boldness. For both sexes, several competing models for breeding success did not include boldness.

Lines 254-255: Please explain why you think carry-over effects of time flying and foraging were similar. If I understood correctly, effects of foraging activity differed between the sexes and the two activities had opposite effects on females.

General comments:

Line 81: In most seabird species, males and females both incubate eggs and rear young, so at least behaviorally, there is little difference in breeding roles. Relative energetic investment in breeding between the sexes is debated.

Lines 298-300: The correct metric here would be lifetime reproductive success. If shy individuals have lower average success than bold individuals, but breed longer, lifetime reproductive success could be equal.

Tables and Figures:

Table 2: It would be helpful to show all models with Δ AICc within 2 (those models that were dropped under the nesting rule).

Tables 2 and 3: Please explain what the gray areas mean. The reader can figure it out, but it would be simpler if you explained it.

All figures: I cannot tell the shades of dots apart. Please use more color contrast and larger dots so the colors show more.

Editorial comments:

Line 22: Delete "of" from "effects of on breeding".

Line 56 & others: Spell out POLS. We don't need more acronyms that just make it harder for nonspecialists to read the paper.

Line 98: "great control" should perhaps be "greater control" as it is in relationship to males' control.

Lines 155-158: The second sentence seems to mostly repeat the first sentence. Either combine the two sentences or reword to make it clear how they differ.

Lines 212-213: Figure 3 is called out before Figure 2. They should be reversed so Figure 2 is called out first.

Line 242: "high flying activity" is a kind of ambiguous term (sounds like flight altitude). Perhaps "spending more time flying" would be clearer.

Line 246: Please spell out pace-of-life syndrome. If you don't spell it out in the rest of the discussion, please do so here, because I already forgot what POLS stands for by the time I got to the discussion.

Lines 307-308: Please delete extra words in this sentence.

Referee: 2

Comments to the Author(s)

The major goal of this research is to test whether individual differences in personality traits (particularly in bold-shy axis) influence the strength of carry-over effects on breeding performance. More specifically, the authors test a hypothesis deriving from pace-of-life syndrome: carry-over effects on breeding performance are weaker in bold (fast-paced) individuals, as they tend to maintain an allocation to reproduction irrespective of previous conditions, while shy (slow-paced) individuals experience stronger carry-over effects because they favor allocation to self-maintenance over current reproduction. They tested this hypothesis in black-legged kittiwakes, which exhibit biparental care behavior. First of all, they discovered that a negative carry-over effect of non-breeding activity on breeding performance. Second, they further revealed that carry-over effects of non-breeding activity on breeding performance were stronger in shyer individuals: active winters were followed by later breeding onset and worse breeding performance in shy birds, but these effects were weaker or undetected in bolder individuals. Overall, this is a great study and I enjoy reading this manuscript, but there are a few places need to be addressed or further clarified in the article. Following are some major concerns and minor suggestions to the authors:

1. Boldness traits in statistical analysis:

According to the authors' description in the manuscript, they assessed boldness traits using a novel object test in 2017 and 2018, and found these scores are highly repeatable within a single breeding season. Here are some of my concerns:

(1) did the authors only measure boldness in 2017 and 2018 and use this boldness score to fit all the linear mixed-effects models? As far as I read, the non-breeding activity (perhaps also breeding performance) was collected between 2012 and 2018, which means the authors used the boldness data from the later time point to predict the influences on breeding performance in earlier times (2012-2016). My question is that how repeatable this boldness index across different breeding seasons? A challenge when studying pace-of-life syndrome is the fact that behavior traits or physiological traits are highly labile, which means they are sensitive to study methods and even to environmental variation (Beckmann and Biro, 2013; Carter et al., 2013; Biro, Adriaenssens and Sampson, 2014). I understand that personality traits like boldness are considered as stable traits, but they still can vary across at different time points. Use the single-year boldness score (e.g., boldness score in 2017-2018) as an independent variable to test its influence on breeding success across several years (e.g., breeding performance in 2012-2016) might not be an ideal setup.

References mentioned here:

Beckmann C and Biro PA (2013). *Ethology*, 119, 937-947.

Carter AJ, Feeney WE, Marshall HH, Cowlshaw G and Heinsohn R (2013). *Biological Reviews*, 88, 465-475.

Biro PA, Adriaenssens B and Sampson P (2014). *Journal of Animal Ecology*, 83, 1186-1195.

(2) The authors mentioned there are 80 individuals were tested in boldness test. Are these the same group of animals sampled for non-breeding activity and scored breeding performance? If yes, how many males and females in this population?

(3) Are all these 5 behaviors representing boldness? Which are the loading weights of each behavior in the PC1? I usually would recommend including a table for principal component analysis output in supplementary data to illustrate the cumulative variance explained in all PCs, just like what the authors did in their earlier publication in 2020 (Harris et al. (2020) *Animal Ecology* 89:68-79).

2. Data interpretation:

The authors indicated that their results revealed predominantly negative effects of winter activity on breeding performance, which I totally agree with it. However, there are some positive carry-over effects on breeding. For example, in males, "flight" activity in the non-breeding season has a positive effect on offspring survival; in females, "forage" activity in non-breeding season also has a positive effect on offspring survival. I think the authors did not further explain these findings. I also think the authors should discuss why these two non-breeding activity traits (forage and flight) sometimes have opposite effects on the same breeding traits. To me, one thing that can be discussed is that, although the authors treated both "forage" and "flight" as the non-breeding activity, they may affect breeding in different ways because the "flight" is an energy-consuming behavior, while "forage" is an energy-refilled behavior. I think that is why we sometimes see the opposite effects of these two behaviors on breeding performance. For instance, in the offspring survival model, forage behavior is positively associated with offspring survival, but flight behavior is negatively associated with offspring survival, though we see both behaviors are negatively correlated with colony arrival date and lay date.

3. Materials & Methods:

The author did not describe how do they access breeding performance and breeding success in this section. There is no clear definition for some important terms, like "colony arrival date" and "lay date". Is the date starting from the first day of each year? Or it is starting from the last day of the previous breeding season? Also, what are the relationships of these two traits (colony arrival date and lay date) and breeding performance? I am a bit confused at the beginning because these

two traits are negatively correlated with breeding success, but offspring survival is positively correlated with breeding success.

4. Results and Tables:

I like the way the authors used Akaike's Information Criterion (AICc) to test and select LMM models! But in the final model (Table 3), I think the authors should also report F-value and p-value for all fixed variables and interaction terms. In the Results section and Tables, there are no p-values reported in context. I understand p-value is not everything, but without it, it is hard to convince the readers that which variables have significant effects on responding variables. In the figures, since the authors use the LMM regression models, it would be better to report the estimate (slope), R²-values, p-values of the best-fitted lines so that it would be easy for readers to understand how the strong/weak the relationships are.

Minor comments:

Line 66-69:

If at the population levels, previous study has shown the fast-paced and slow-paced individuals are responding differently to stress in reproductive behavior, what is the advantage of doing similar experiment at individual levels? I think the authors can further elaborate this more to emphasize the importance of their research.

Line 95:

But in the present setup, the authors cannot compare the sex differences because (1) they used different model components in males versus in females; (2) they analyze males and females separately.

Line 110-111:

The authors should also include the protocol of how they conduct blood collecting and DNA extraction for identifying sex in the Methods.

Line 159:

How many males and females in these 49 individuals? Do all these individuals have boldness data from each year? Please also see the first point in the major concerns.

Line 182-185:

It seems that "arrival date" and "lay date" are associated with one of each other. Is there any multi-collinearity issue in this model?

Line 210:

I understand that sex variable can be separately analyzed in "colony arrival" and "lay date" models, but "offspring survival" are contributed by both males and females because this species exhibit biparental care (as the authors mentioned in previously), and therefore separately analyzing sex might raise some issues since you wouldn't be able to count the interactions between the male and female.

Line 212:

Please also report statistical results (e.g., F-value and p-value) of each result to support the statement. Please also see the fourth point in the major concerns.

Line 213:

Please specify whether it is a positive or negative association.

Line 216:

But in both males and females, the authors also find positive carry-on effects on offspring survival. Please also see the second comment in the major concerns.

Line 225-231:

Do all these effects reach statically significant? I understand p-value is not everything, but if it is applicable, the authors should report the statistical results to back up their statement.

Line 244-245:

Not exactly true. In the offspring survival model, the "flight" and "forage" traits have opposite effects on offspring survival.

Line 245:

Any statistical analysis can support this statement?

Line 287-288:

Which part of data can support this statement?

Line 322-324:

Did the author have "size" data to incorporate into their analysis?

Reference:

Please add reference numbers into the reference list. Otherwise, it is difficult to track the citations which are mentioned in the context.

Line 338: "fo 67:1-48"

Is "fo" a typo?

Tables:

Did all analysis include "bird identity" and "year" as random variables? If yes, please specify in the table legend.

Table 1 and 2

The "X" and "-" is confusing. The authors can leave the non-included variable as blank.

Also, I would recommend moved Table 1 and 2 to supplementary data.

Moreover, in addition to estimate and importance, the authors should also report F-values and p-values (if applicable) of each variable. Please also see the fourth comment in the major concerns.

Figures:

Please also report slope of the best-fitted line, R2-values, F-values and p-values in either the figure or in the context.

Author's Response to Decision Letter for (RSPB-2020-0324.R0)

See Appendix A.

RSPB-2020-2381.R0

Review form: Reviewer 1

Recommendation

Accept with minor revision (please list in comments)

Scientific importance: Is the manuscript an original and important contribution to its field?
Excellent

General interest: Is the paper of sufficient general interest?
Good

Quality of the paper: Is the overall quality of the paper suitable?
Excellent

Is the length of the paper justified?
Yes

Should the paper be seen by a specialist statistical reviewer?
Yes

Do you have any concerns about statistical analyses in this paper? If so, please specify them explicitly in your report.
No

It is a condition of publication that authors make their supporting data, code and materials available - either as supplementary material or hosted in an external repository. Please rate, if applicable, the supporting data on the following criteria.

Is it accessible?
No

Is it clear?
N/A

Is it adequate?
N/A

Do you have any ethical concerns with this paper?
No

Comments to the Author

I find the manuscript to be improved and close to ready for publication. I think some editing, a little re-wording, and still some better explanations are all that are required. I agree that you should not use both p-values and AIC.

Line 55: Why not just say "Among individuals, variation ..." instead of "At the among-individual level, variation ..."?

Line 56: Missing the closing parenthesis.

Line 60: I think you mean "reproduction", not "reproductive".

Line 116: Add "see supplementary material", which you say for all the other appendices.

Lines 134-136: I still think this needs more explanation. Please state exactly what result from the linear model is the single estimate of boldness per individual. You addressed this in your response to reviewers, but did not clarify it in the manuscript.

Line 158: Delete "for": "except in cases".

Lines 180-183: Spell out "linear mixed effects models" on line 180 instead of line 183.

Line 184: Were bird identity and year nested?

Line 189: I think the second “interval” should be “interaction”.

Lines 233-238: Refer to Table B1 here. I really think this table should be in the paper, not in the supplementary file. It looks like bold individuals sat tight and very shy individuals flew away. The other behaviors did not contribute much. It would be worth a sentence about the most important “bold” and “shy” behaviors.

Line 392: Delete one “that”.

Line 399: Delete the second “than”.

Table 1: I think “estimated” in the first line of the caption should be “estimates”. Please re-word the second sentence. It makes no sense. Please refer to Table E1 at the end of the caption.

Supplementary files:

Appendix A: I would put this in the paper. It is only two paragraphs.

Appendix B: I think this should be in the paper.

Appendices C, E, and F: These tables are very helpful.

Appendix D: Change “Figure C1” to “Figure D1”.

I’m not sure, but these colors might be problematic for people with some kinds of color blindness. Please check this. Otherwise, I like this figure.

Appendix E: You numbered citations in Appendices A and C, but not here. Please be consistent.

Appendix G:

Line 2 of the caption: “between 0-5 years” does not work. I think “ranged from 0 to 5 years” would be better.

You refer to Table 2 of the main paper, but there is only 1 table in the main paper now.

Change “Table F1 below” to “Table G1 below”.

You reversed the estimates for “Boldness x foraging” and “Boldness x flight” in Table 1, but not here.

Review form: Reviewer 2

Recommendation

Accept with minor revision (please list in comments)

Scientific importance: Is the manuscript an original and important contribution to its field?

Excellent

General interest: Is the paper of sufficient general interest?

Good

Quality of the paper: Is the overall quality of the paper suitable?

Good

Is the length of the paper justified?

Yes

Should the paper be seen by a specialist statistical reviewer?

No

Do you have any concerns about statistical analyses in this paper? If so, please specify them explicitly in your report.

No

It is a condition of publication that authors make their supporting data, code and materials available - either as supplementary material or hosted in an external repository. Please rate, if applicable, the supporting data on the following criteria.

Is it accessible?

Yes

Is it clear?

Yes

Is it adequate?

Yes

Do you have any ethical concerns with this paper?

No

Comments to the Author

I have carefully read this resubmitted manuscript, and I think this manuscript had great improvements. I also think the authors have thoroughly addressed most of my comments in this revised manuscript. I only have a few additional comments for the authors.

Line 51: change "...by high allocation to current breeding and low survival" to "...by high allocation to current breeding but low survival".

Line164-168: Are there any differences in the length of non-breeding season between shy and bold individuals? I am wondering if the "length of the non-breeding season" would be a better response variable than "colony arrival date", because it represents how much time an individual spent in preparing for the next breeding season, which can reflect its actual physical condition.

Line 182-183 and other models: Is there any transformation required for the variables to meet the normality assumption in LMMs? If yes, please specify these variables and describe how they are transformed.

Line 198: Any reference for setting up this criterion (i.e., < 2 AIC units)?

Line 217: please specify what the statistic test is here.

Line 229: I think this should be Appendix G, not F.

Line 244: There are no Table D1 in the Appendix. Do the authors mean Table C1 here?

Line 248-249, Line 314 and Figure 1: In this paragraph and several other places, the authors mentioned that negative carry-over effects were strongest in shy individuals, but how do they determine the effect is strong or weak? I cannot tell the level of significance between shy and bold individuals from Table1, and there were no statistical values reported in Figure 1.

In Figure 1g, it seems that there are negative carry-over effects in both shy and bold individuals, did the authors determine the strength of carry-over effects by slopes or R-square value?

Also, in Figure 1, the authors mentioned that estimates are presented for the boldest individuals (+1 standard deviation) and shyest individual (-1 standard deviation); I wonder how many individuals are categorized in boldest and shyest groups? Can they represent most of the bold and shy individuals in the population? I am worried about if the results might be driven by few individuals with extremely behavioral phenotypes (e.g., extremely bold or shy individuals).

Line 253-255: I am wondering if "lay date models" are important in males because the lay date is majorly controlled by the female itself, and therefore this trait may not be directly related to male reproduction. Do you think this might be the reason why the authors did not see the effects of the interaction term (boldness x activities) on these traits? If the authors used the behavioral traits which is more relevant to male reproduction (e.g., first courting date or first copulation date) in these models, would they expect to see a different result?

Line 388-390: Please add references to support this sentence.

Line 399: remove extra "than" in this sentence.

Line 400: I am still interested to see if the authors pool male and female as one "sex" variable and include it in the "offspring survival model", would they see a different result? The offspring survival should account for both males' and females' investment. Therefore, analyzing males and females separately might raise some issues since they are unable to count the interactions between both sexes. I understand the three-way interactions (sex, boldness, and activity) may make this model over complex. Still, it may not necessarily need to be included in the models depending on its importance.

Decision letter (RSPB-2020-2381.R0)

23-Oct-2020

Dear Dr Harris:

Your manuscript has now been peer reviewed and the reviews have been assessed by an Associate Editor. The reviewers' comments (not including confidential comments to the Editor) and the comments from the Associate Editor are included at the end of this email for your reference. As you will see, the reviewers and the Editors have raised some concerns with your manuscript and we would like to invite you to revise your manuscript to address them.

When submitting your revision please upload a file under "Response to Referees" in the "File Upload" section. This should document, point by point, how you have responded to the

reviewers' and Editors' comments, and the adjustments you have made to the manuscript. We require a copy of the manuscript with revisions made since the previous version marked as 'tracked changes' to be included in the 'response to referees' document.

Research ethics:

Use of animals and field studies:

It is a condition of publication that you make available the data and research materials supporting the results in the article (<https://royalsociety.org/journals/authors/author-guidelines/#data>). Datasets should be deposited in an appropriate publicly available repository and details of the associated accession number, link or DOI to the datasets must be included in the Data Accessibility section of the article (<https://royalsociety.org/journals/ethics-policies/data-sharing-mining/>). Reference(s) to datasets should also be included in the reference list of the article with DOIs (where available).

Online supplementary material will also carry the title and description provided during submission, so please ensure these are accurate and informative. Note that the Royal Society will not edit or typeset supplementary material and it will be hosted as provided. Please ensure that

the supplementary material includes the paper details (authors, title, journal name, article DOI). Your article DOI will be 10.1098/rspb.[paper ID in form xxxx.xxxx e.g. 10.1098/rspb.2016.0049].

Please submit a copy of your revised paper within three weeks. If we do not hear from you within this time your manuscript will be rejected. If you are unable to meet this deadline please let us know as soon as possible, as we may be able to grant a short extension.

Best wishes,
Dr Maurine Neiman
mailto:proceedingsb@royalsociety.org

Associate Editor Board Member

Comments to Author:

Thank you for addressing the referees comments so thoroughly, and my own. The referees have some remaining queries about the analyses (e.g. personality data distribution, relevance of sex-specific models) and otherwise suggest a few clarifications.

Reviewer(s)' Comments to Author:

Referee: 1

Comments to the Author(s).

I find the manuscript to be improved and close to ready for publication. I think some editing, a little re-wording, and still some better explanations are all that are required. I agree that you should not use both p-values and AIC.

Line 55: Why not just say "Among individuals, variation ..." instead of "At the among-individual level, variation ..."?

Line 56: Missing the closing parenthesis.

Line 60: I think you mean "reproduction", not "reproductive".

Line 116: Add "see supplementary material", which you say for all the other appendices.

Lines 134-136: I still think this needs more explanation. Please state exactly what result from the linear model is the single estimate of boldness per individual. You addressed this in your response to reviewers, but did not clarify it in the manuscript.

Line 158: Delete "for": "except in cases".

Lines 180-183: Spell out "linear mixed effects models" on line 180 instead of line 183.

Line 184: Were bird identity and year nested?

Line 189: I think the second "interval" should be "interaction".

Lines 233-238: Refer to Table B1 here. I really think this table should be in the paper, not in the supplementary file. It looks like bold individuals sat tight and very shy individuals flew away. The other behaviors did not contribute much. It would be worth a sentence about the most important "bold" and "shy" behaviors.

Line 392: Delete one "that".

Line 399: Delete the second “than”.

Table 1: I think “estimated” in the first line of the caption should be “estimates”. Please re-word the second sentence. It makes no sense. Please refer to Table E1 at the end of the caption.

Supplementary files:

Appendix A: I would put this in the paper. It is only two paragraphs.

Appendix B: I think this should be in the paper.

Appendices C, E, and F: These tables are very helpful.

Appendix D: Change “Figure C1” to “Figure D1”.

I’m not sure, but these colors might be problematic for people with some kinds of color blindness. Please check this. Otherwise, I like this figure.

Appendix E: You numbered citations in Appendices A and C, but not here. Please be consistent.

Appendix G:

Line 2 of the caption: “between 0-5 years” does not work. I think “ranged from 0 to 5 years” would be better.

You refer to Table 2 of the main paper, but there is only 1 table in the main paper now.

Change “Table F1 below” to “Table G1 below”.

You reversed the estimates for “Boldness x foraging” and “Boldness x flight” in Table 1, but not here.

Referee: 2

Comments to the Author(s).

I have carefully read this resubmitted manuscript, and I think this manuscript had great improvements. I also think the authors have thoroughly addressed most of my comments in this revised manuscript. I only have a few additional comments for the authors.

Line 51: change “...by high allocation to current breeding and low survival” to “...by high allocation to current breeding but low survival”.

Line 164-168: Are there any differences in the length of non-breeding season between shy and bold individuals? I am wondering if the “length of the non-breeding season” would be a better response variable than “colony arrival date”, because it represents how much time an individual spent in preparing for the next breeding season, which can reflect its actual physical condition.

Line 182-183 and other models: Is there any transformation required for the variables to meet the normality assumption in LMMs? If yes, please specify these variables and describe how they are transformed.

Line 198: Any reference for setting up this criterion (i.e., < 2 AIC units)?

Line 217: please specify what the statistic test is here.

Line 229: I think this should be Appendix G, not F.

Line 244: There are no Table D1 in the Appendix. Do the authors mean Table C1 here?

Line 248-249, Line 314 and Figure 1: In this paragraph and several other places, the authors mentioned that negative carry-over effects were strongest in shy individuals, but how do they

determine the effect is strong or weak? I cannot tell the level of significance between shy and bold individuals from Table 1, and there were no statistical values reported in Figure 1.

In Figure 1g, it seems that there are negative carry-over effects in both shy and bold individuals, did the authors determine the strength of carry-over effects by slopes or R-square value?

Also, in Figure 1, the authors mentioned that estimates are presented for the boldest individuals (+1 standard deviation) and shyest individual (-1 standard deviation); I wonder how many individuals are categorized in boldest and shyest groups? Can they represent most of the bold and shy individuals in the population? I am worried about if the results might be driven by few individuals with extremely behavioral phenotypes (e.g., extremely bold or shy individuals).

Line 253-255: I am wondering if "lay date models" are important in males because the lay date is majorly controlled by the female itself, and therefore this trait may not be directly related to male reproduction. Do you think this might be the reason why the authors did not see the effects of the interaction term (boldness x activities) on these traits? If the authors used the behavioral traits which is more relevant to male reproduction (e.g., first courting date or first copulation date) in these models, would they expect to see a different result?

Line 388-390: Please add references to support this sentence.

Line 399: remove extra "than" in this sentence.

Line 400: I am still interested to see if the authors pool male and female as one "sex" variable and include it in the "offspring survival model", would they see a different result? The offspring survival should account for both males' and females' investment. Therefore, analyzing males and females separately might raise some issues since they are unable to count the interactions between both sexes. I understand the three-way interactions (sex, boldness, and activity) may make this model over complex. Still, it may not necessarily need to be included in the models depending on its importance.

Author's Response to Decision Letter for (RSPB-2020-2381.R0)

See Appendix B.

Decision letter (RSPB-2020-2381.R1)

12-Nov-2020

Dear Dr Harris

I am pleased to inform you that your manuscript entitled "Personality-specific carry-over effects on breeding" has been accepted for publication in Proceedings B.

Open Access

Paper charges

Sincerely,

Dr Maurine Neiman

Associate Editor:

Board Member

Comments to Author:

(There are no comments.)

Appendix A

09-Apr-2020

Dear Ms Harris:

I am writing to inform you that your manuscript RSPB-2020-0324 entitled "Personality-specific carry-over effects on breeding" has, in its current form, been rejected for publication in Proceedings B.

This action has been taken on the advice of referees, who have recommended that substantial revisions are necessary. With this in mind we would be happy to consider a resubmission, provided the comments of the referees are fully addressed. However please note that this is not a provisional acceptance.

Sincerely,

Dr Maurine Neiman
mailto: proceedingsb@royalsociety.org

Dear Dr Neiman,

We are very grateful for the editorial and reviewer comments on our manuscript, which we found very insightful. We hope that we have made substantial improvements to the manuscript based upon these suggestions. Please see below for our specific responses; please note that our line numbers below reference the clean copy of the new version of our manuscript.

Associate Editor
Comments to Author:

This is a neat combination of large-scale field datasets, and a pleasure to read. The carry-over effects are interesting, even before the contribution of personality. I agree with the referees that areas need clarification in order to fully assess the manuscript. If methodological points of referee 1 can be addressed, then I find the links between over-winter behaviour and reproduction compelling. However, the key aspect is personality. And I share the concern raised by Referee 2 on the timing of boldness assays.

We are glad to hear you enjoyed the manuscript, and thank you for your very helpful comments!

While the emphasis of the manuscript is directional (personality mediates carry-over effects), the over-winter data appear to be collected up to 6 years before the boldness tests. First, an alternative explanation is therefore carry-over effects of over-winter behaviour on both boldness and reproduction. Indeed, it would be really interesting if this were the case. While the lack of correlation in the first model suggests it's not, interval (between over-winter and boldness measurements) should be controlled for in this model. If data allow, a stronger option would be to explicitly correlate over-winter behaviour in the 2017-2018 winter to change in boldness between these years.

We acknowledge that in an ideal situation, we would have conducted boldness tests over the entire 6-year study period over which winter data was collected. However, this study was designed to add personality assays to pre-existing tracking data, which as you point out below is important (particularly in biologging studies owing to ethical, logistical and financial constraints on the number of animals equipped with loggers). Such data collection regimes are often carried out when substantial pre-existing data is paired with new data designed to test novel hypotheses. In this case, the archive of foraging data provides a unique opportunity to examine the importance of personality.

We feel that the assumption that boldness scores are consistent across 5 years is valid, and not unusual in the literature (note that while non-breeding data were collected over 6 years, the maximum interval between collection of winter behaviour data and an individual's first boldness test is 5 years). We believe the assumption is valid given that, as you note, boldness is highly repeatable over the 2-year test period ($R \pm S.E. = 0.61 \pm 0.07$), with virtually no reduction compared to that reported in a single year ($R \pm S.E. = 0.68 \pm 0.07$; Harris *et al.* 2020, *J. Anim. Ecol.* 89:68), and in other seabird species boldness has been shown to be repeatable over periods of at least 5 years (Patrick & Weimerskirch 2013, *Ecol. Evol.* 3:4291; Patrick *et al.* 2017, *J. Anim. Ecol.* 86:1257). More broadly, boldness is widely considered to be arepeatable and heritable trait in many species (Dochtermann *et al.* 2015, *Proc. B.* 282:20142201; see also Patrick & Weimerskirch 2013 for heritability in seabirds specifically). Accordingly, we believe our inference that boldness scores as collected over two years are representative of birds' behaviour across the study period is appropriate.

Exploring carry-over effects on boldness, as you suggest, would be an interesting addition to the growing research interest in plasticity of personality traits (i.e. changes in personality in response to the environment; research also highlighted by Reviewer 2). Personality and plasticity are two separate components of individual behavioural variation: personality is the "intercept" value for an individual, which is fixed over time, while plasticity in this trait is the "slope" in response to changes in the environment (or here, winter foraging conditions). Investigating plasticity in personality in response to carry-over effects would be extremely interesting, but we feel it is outside the scope of this paper. This would also require more than 2 years of boldness measurements to quantify plasticity, ideally more than 5 years to fit these models correctly, using random slopes to measure the change in boldness (following Dingemanse *et al.* 2010, *TREE* 25:81). However, we do agree it is important to test whether potential carry-over effects on boldness could be affecting our analyses, and we did so following both of your suggested approaches.

Firstly, we added the interval between winter behaviour and boldness measurements to our models examining the relationship between winter behaviour and boldness. If winter behaviour does have carry-over effects on boldness, we would expect to find a relationship between boldness and winter behaviour when the interval is controlled for. For both winter flight and winter foraging behaviour, we found no evidence of an effect of interval, with the best model containing the intercept only, supporting that results are not driven by concomitant carry-over effects on both boldness and breeding. We have included these results in our supplementary materials Appendix C, and reported this in the text after making clearer the structure of the data in terms of the interval between boldness test and winter data:

New text at lines 185-190: "Because boldness tests were conducted exclusively during the final two years of tracking (2017 and 2018), there was an interval of 0-5 years between the collection of non-breeding data and boldness data (mean interval: 1.79). We therefore

controlled for the interval between non-breeding period and an individual's first boldness test in these models, and found no support for an effect of interval or the interval between interval and boldness on non-breeding activity (supplementary material Appendix C)."

Secondly, for the individuals which were tested in both 2017 and 2018, we also followed your suggestion to look at whether changes in boldness between years were correlated with non-breeding activity. We found no evidence of a correlation between change in yearly boldness scores and non-breeding foraging ($R = -0.011$, $p = 0.963$) nor non-breeding flight ($R = -0.049$, $p = 0.829$). We therefore infer that our findings are not a result of carry-over effects on boldness as well as breeding. At present we have not included these analyses in the manuscript at present, as we feel data from two years is insufficient to look at the change in boldness, but we have added a paragraph to the discussion to raise that this is a possibility:

New text at lines 361-372: "Carry-over effects may also interact with boldness by acting upon personality traits directly. Personality traits are typically characterised by their stability, but recent work has recognised the importance of within-individual changes in personality in response to environmental conditions, known as behavioural plasticity [62,63]. Our method of assaying boldness captures individuals' propensity to defend their nest, and we may therefore expect that when carry-over effects of winter conditions lead an individual to invest less in reproductive performance, they should also behave more shyly. By assaying boldness in individuals over periods of several more years, it would be possible to quantify individuals' plasticity in personality in relation to non-breeding conditions, and test whether carry-over effects also act upon personality traits. Furthermore, using longitudinal boldness data, future work could test whether individuals consistently differ in their plasticity in response to winter conditions [62,64], and examine whether plasticity in personality is adaptive, and its consequences for lifetime fitness."

Second, the high over-year repeatability is remarkable – clearly a striking behavioural difference. But I am concerned as to whether this is evidence enough of consistency for the period of study, where 6 years is a large part of the average post-recruitment lifespan. For example, are age-related shifts in reproductive investment described in kittiwakes? It is possible that repeatability in boldness, which is effectively investment in nest defence, is inflated by comparing adjacent years within relatively young versus old individuals. This becomes an issue if long intervals are prevalent in the dataset – which may not be the case - as there is then a risk that relatively young (less efficient?) or middle aged (peak performance?) winter strategies are being correlated to characteristically 'middle-aged' versus 'old' personality types. Adding interval to analyses and/or re-running with a conservative subset of the data that have a short interval between measurements would be useful.

This raises a good point regarding the short- versus long-term repeatability of animal behaviour. Our understanding of personality from other systems gives good reason to interpret personality scores as indicative of phenotypic behavioural differences that persist over individuals' lifetimes, as indeed many studies examining personality in wild animals do. Studies specifically investigating age effects on boldness (including in other bird species) have found changes in boldness with age to be very minor when compared to differences among individuals of the same age class (e.g. Patrick & Weimerskirch 2015, *Proc. B.* 282:20141649, Holtmann *et al.* 2017, *Proc. B.* 284:20170943). Regarding age effects in the kittiwakes, age is not known for most individuals in this study population. However, as you suggest this would only be a substantial issue if there are many long intervals between tracking data and boldness tests in the dataset. In Figure R1 (below), we show the distribution of the intervals. The mean interval was 1.79, and as Figure R1 shows, in the majority of cases individuals were tested for boldness the same year that activity loggers were recovered, and the longest interval of five years occurred only in three cases. We therefore feel the length of interval is unlikely to have bearing on our results.

Figure R1. Histogram of the time interval between winter behaviour and boldness measurements, in years.

We also followed your suggestion and reran carry-over effects models with a conservative subset of the data, where the interval between measurements was two years or less. For each analysis (e.g. for effects of colony arrival date in males, then in females, etc.) we fitted the top-ranking models (those presented in Table E1 of the supplementary materials) on the subset of data. We compared the parameter estimates from these models to the estimates presented in Table 1 of the main paper, to see whether results differed substantially in strength and direction. We present both sets of estimates \pm standard error for all variables in initial models in Table R1 below. Parameter estimates from the subset data matched those from the full dataset well in terms of both direction and strength: there are a couple of exceptions, for example the effect of boldness on offspring survival of males is stronger when we subset the data, although the direction of the relationship remained the same. We therefore interpret this as strong evidence that our findings are not driven by the intervals between boldness tests and non-breeding tracking in our dataset. We have now also added this subset analysis to the supplementary materials (Appendix G), to show readers that our findings are supported even when removing long intervals in the data.

Table R1. Estimates and standard errors for all retained variables, as run on a conservative subset of data where the interval between boldness and winter behaviour measurements was 2 years or less.

Response	Predictor	Males		Females	
		Full data	Subset data	Full data	Subset data
Colony arrival date	Intercept	118.41 \pm 2.30	117.29 \pm 1.77	119.16 \pm 1.33	117.95 \pm 1.70
	Boldness	0.28 \pm 0.99	0.17 \pm 1.31	1.97 \pm 0.94	2.28 \pm 1.01
	Foraging	2.12 \pm 0.95	2.15 \pm 1.09		
	Flight	0.00 \pm 1.04	-0.49 \pm 1.50	2.48 \pm 0.88	1.76 \pm 1.08
	Boldness x foraging	-2.06 \pm 1.10	-1.97 \pm 1.40		
	Boldness x flight	-2.15 \pm 1.03	-2.00 \pm 1.67		
Lay date	Intercept	162.94 \pm 1.33	160.89 \pm 1.17	161.63 \pm 1.33	160.65 \pm 1.78
	Boldness			2.76 \pm 0.63	2.36 \pm 0.94
	Foraging	1.97 \pm 0.95	1.76 \pm 1.69		
	Flight	1.40 \pm 0.95	0.15 \pm 1.43	2.96 \pm 0.79	2.45 \pm 1.40
	Boldness x foraging				
	Boldness x flight			-1.77 \pm 0.62	-1.46 \pm 1.13
Offspring survival	Intercept	13.02 \pm 2.95	13.53 \pm 5.10	14.52 \pm 3.62	14.51 \pm 5.44
	Boldness	-0.36 \pm 1.23	-1.86 \pm 1.78	-1.75 \pm 1.37	-1.88 \pm 1.77
	Foraging	-2.06 \pm 1.32	-2.15 \pm 1.74	1.50 \pm 1.32	2.38 \pm 1.80
	Flight	1.39 \pm 1.44	1.83 \pm 2.01	-1.02 \pm 1.50	-0.54 \pm 2.02
	Boldness x foraging	2.13 \pm 1.32	2.06 \pm 1.71		
	Boldness x flight				
	Arrival date			-1.10 \pm 1.52	-1.90 \pm 2.09
	Lay date	-1.31 \pm 1.41	-1.24 \pm 1.70	-1.35 \pm 1.50	-1.26 \pm 1.94

These concerns may be simply addressed by explaining the data distribution, which is not quite clear in the manuscript. Specifically, the number of birds with over-year replication in the boldness test (lines 120 and 200), the average number of over-winter measurements per bird (range 1-4 on lines 136/159) and, importantly, average/range of interval. As a ringed population, is age or size known? We have now added more information to the manuscript regarding sample sizes and data distributions, following your suggestions:

- 1) The number of birds with boldness tests in two years:
New text at line 127: “27 individuals were tested in both 2017 and 2018.”
- 2) The average number of over-winter measurements per bird:
New text at line 177: “... a mean of two bird-years per individuals (range 1-5 years).”
- 3) The average and range of the interval between boldness tests and over-winter measurements:
New text at lines 185-187: “Because boldness tests were conducted exclusively during the final two years of tracking (2017 and 2018), there was an interval of 0-5 years between the collection of non-breeding data and boldness data (mean interval: 1.79).”

Unfortunately, age is unknown in this population, and there are no reliable methods of ageing adult kittiwakes in the field based on plumage or size. However we have now added a paragraph to the discussion, discussing the potential for age effects in our findings:

New text at lines 349-359: “Carry-over effects have also been found to vary with age in some species [11]. In wandering albatrosses (*Diomedea exulans*), while younger birds all bred successfully, high foraging effort was linked to increased breeding failure in older individuals [11]. Age differences may thus also explain variation in carry-over effects in our study, although we were unable to explore this possibility because birds in this population were of unknown age. While personality traits have been found to be stable over long periods [e.g. 64], directional changes in boldness have also been documented in some species. Thus, the relationship between boldness and carry-over effects in kittiwakes could be linked to age differences if kittiwakes become shyer in older age. Theory and empirical findings generally support the opposite pattern, whereby animals get bolder with age, as their residual reproductive value decreases [65–67]. Nevertheless, further research should investigate the relationship between age, boldness, and the strength of carry-over effects on breeding.”

It is vitally important to make use of opportunities to link existing datasets and explore longer term patterns. But then limitations that arise should be discussed clearly throughout. If it is not possible to address these concerns with the existing data, I would recommend revising the manuscript to give alternative explanations some or potentially equal weight.

We agree with the editor on the importance of using existing datasets and fully acknowledging the limitations of data and alternative potential explanations for findings. Following your suggestion, we have added new text to the discussion to provide alternative explanations for our findings, including the potential for age-related variation in carry-over effects (lines 349-359; text pasted in response to the comment above) the possibility that carry-over effects may actually act upon personality (lines 361-372), as you suggested in an earlier comment.

Overall, I agree with the referees that this is a thought-provoking study, but further information or analyses are required to evaluate the results. I hope the referees’ clear and supportive comments will be useful in revision. One to note is that both indicate uncertainty over the strength of effects – Referee 1 in terms of the clarity of figures and Referee 2 the goodness of fit. I had a similar sense, as it is not clear in Figures 1a, 1c and Figure 3 how the underlying datapoints are described by the fitted lines (particularly where they extend into areas without data). Thus, it may be necessary to consider how best to present the results.

We hope we have sufficiently provided further information to address yours and the reviewers’ comments. We have spent some time adjusting the figures. Firstly, because these figures were very large, we have arranged them into a single figure, enabling readers to compare all effects more easily. In response to Review 1’s comment about the figures, we have increased point sizes and used a colour

scale with greater contrast to better visualise the results. Following your comment, we also adjusted the fitted lines so that they no longer extend beyond data points – this was an issue, as you pointed out, in the figures where separate lines are plotted for bold and shy individuals, because we had not specified that lines for shy individuals should extend only to where we have data points for shy individuals (and likewise for bold individuals). In addition, previous plots included standard errors, but on reconsideration we have now plotted confidence intervals as we believe these are more appropriate. In response to Reviewer 2's comments, we have now reported R^2 values (conditional and marginal). These were added to the previous Table 2, but following Reviewer 2's recommendation we have since moved this table to the Supplementary materials (please see Table E1). These coefficients of variations enable readers to better evaluate goodness of fit, and we also discuss their meaning in terms of the variation explained by fixed effects in the Results section.

New text at lines 265-268: "Coefficients of determination indicated that the variation explained by fixed effects was between 8-22% for colony arrival date, 8-36% for lay date, and 1-12% for offspring survival (see supplementary materials, Table E1). This suggests that most of the variation in breeding was explained by differences among individuals and years, particularly in offspring survival models."

Reviewer(s)' Comments to Author:

Referee: 1

Comments to the Author(s)

This study tested for sex- and personality-specific carry-over effects from winter to the following breeding season in black-legged kittiwakes. It found, as predicted, that carry-over effects (timing of breeding and breeding success) were stronger in shy individuals than in bold individuals. It also found opposite effects in males and females.

The paper is well written, except for a few typos. I could not evaluate the use of the literature because the citations in the text are numbered but the reference list is in alphabetical order without numbers. Thanks very much for your positive and helpful comments. We have now amended the reference list to include numbers.

I have a few questions and concerns about the methods and interpretation of the results:

Lines 124-125: Adjusted repeatability how and why?

Adjusted repeatability estimates the variance explained by differences between individuals after accounting for other factors which might influence the variable being measured (from Nakagawa & Schielzeth 2010, *Biological Reviews* 85:935-956). Here, we control for test date, breeding stage and test number, by fitting these variables as fixed effects. We have added text to explain this:

New text at lines 130-131: "We measured adjusted repeatability (repeatability after controlling for confounding effects [38])"

Lines 127-128: I don't follow this. What exactly was used from the regression as the measure of boldness? Why is individual ID a fixed effect and not a random effect? Does this model allow for individual slopes? How does it account for differences in the number of times individuals were tested?

As single measures of boldness per individual, we extracted parameter estimates for each level of the individual ID fixed effect in a linear model, to use as a single estimate of boldness per individual. We chose this method because parameter estimates are considered better estimates of individual behaviour than individual point estimates from random effect in mixed models (also known as best linear unbiased predictors, or BLUPs; Hadfield *et al.* 2010, *Am. Nat.* 175:116-125). BLUPs are predicted with large amounts of error, and their use in secondary analyses is therefore regarded as anticonservative (Houslay & Wilson 2017, *Behav. Ecol.* 28:948). Our use of parameter estimates instead follows other examples (e.g. Quinn *et al.* 2009, *J. Anim. Ecol.* 78:1203-1215; Patrick &

Weimerskirch 2014: *PLoS ONE* 9:e87269). This method does not allow for individual slopes. In terms of the number of times individuals were tested, we also fitted test number (i.e. an individual's 1st test, 2nd test, ...) as a fixed effect, such that individual parameter estimates are adjusted for the number of times and individual had previously been tested. To make this clearer, we have elaborated on the meaning of “test number” in the methods:

New text at line 133: “test number (the number of times an individual had previously been tested).”

We have also added the citations above as references to other studies applying this method to extract single estimates of individual behaviour from linear models (to line 134).

Lines 139-141: This could be biased if short flights or immersions were missed by the C65 tags. Does it change anything if the data from the MK4083 tags are subsampled at 30-second intervals? If 30 seconds is shorter than any activity bout, state that.

The immersion loggers we used take wet-dry readings every 3-seconds (MK4083) or 30-seconds (C65), but both loggers do not store these raw wet-dry readings, instead storing only the number of wet readings recorded in a 10-minute window. Thus, data from MK4083 loggers ranges from 0-200 (because there are 200 x 3-second readings in a 10-minute window), while data C65 loggers ranges from 0-20 (because there are 20 x 30-second readings in a 10-minute window). Because of the way the loggers store the data, it is not possible to strictly subsample the data by taking every 10th reading from the MK4083 loggers in order to fully replicate the sampling methods of the C65 data. This is why we instead divided the total number of wet fixes from MK4083 by 10 (and round to the nearest integer), so that data from both logger types ranges from 0 (completely dry) to 20 (completely wet). For example, if for a given 10-minute period a MK4083 logger records 43 wet readings, we convert this to the C65 equivalent by dividing by 10 to 4.3, and rounding down to 4. C65 loggers could still technically miss very short bouts of behaviour, but given the high energetic costs to taking off and landing from water in seabirds (e.g. Shaffer *et al.* 2001, *J. Anim. Ecol.* 70:864-874) we would expect such rapid shifts between flying, resting and foraging to be infrequent. We have acknowledged this in the text:

New text at lines 160-163: “Loggers could miss bouts of behaviour shorter than 30 seconds in duration, but we expect such rapid shifts between flying, resting and foraging to be infrequent given the high energetic costs of taking off and landing from water in birds [45,46].”

Lines 164-165: Are boldness and sex correlated? You state that they are not (at least not enough to affect the statistics) at the end of this section, but it would help to state it here, so the reader isn't distracted by wondering about it while reading the rest of the methods.

We have now stated earlier on that we find no difference in boldness between the sexes. We added this to the end of section 2.2 (methods of boldness testing), as it seemed most appropriate to do so here.

New text at line 136-137: “We found no difference in boldness between the sexes (results from a linear model testing for a sex effect on boldness: $p = 0.19$).”

Line 171: I'm not familiar with this nesting rule. It seems that you don't have that many predictors and they are potentially important ones. How would results change if you didn't apply the nesting rule?

When evaluating models by AIC, we rank all models by increasing AIC value, and refine a “top model set” where ΔAIC is less than 2 (Burnham & Anderson 2002). Applying this cut off enables interpretation of the best and most parsimonious models, but AIC is known to favour overly complex models, and simulations show that add a nuisance (completely random) variable predictor to a well-fitting model can lead to its inclusion in the top model set (Arnold 2010, *J. Wildlife Manag.* 74:1175). A conservative approach to improve model inference can therefore be greatly improved by eliminating models from the top set that are more complex versions of simpler (nested) models with better AIC support (Harrison *et al.* 2018, *PeerJ* 6:e4794). For example, a model containing fixed effect A is nested within a model containing effects A + B. Given that (1) AIC is equal to deviance plus the number of parameters x 2, and (2) the addition of a variable to a given model will act to

decrease the deviance, model A + B necessarily cannot be more than 2 AIC units large than model A, even if variable B has null explanatory power. Therefore, even if the model A + B is within 2 AIC units of the first, it should be eliminated. Where predictors are not included in our top model sets, this therefore suggests they did not add sufficient explanatory power to perform better than the retained models, despite that intuitively these may seem to be important predictors. We have altered the methods section slightly here to try to make this approach clearer:

Adjusted text at lines 194-199: “Because AIC can favour overly complex models [47], inference can be improved by eliminating models from the top model set if they are more complex versions of simpler (nested) models with lower AIC_C values, known as the “nesting rule” [48–50]. We therefore applied the nesting rule to prevent the retention of overly complex models, such that when two nested models differed by less than 2 AIC units (indicating that the additional predictor has a very low explanatory power), the simplest model was preferred.”

In addition, as per your later suggestion (five comments down) we have now added a full results table including models eliminated by the nesting rule to the supplementary materials (Appendix F).

Line 181: Time in flight and time foraging should be negatively correlated, as they are mutually exclusive activities. Only if they are each small proportions of total time would they not be correlated. Please expand on this.

Time in flight and time foraging are not mutually exclusive because there is a third behavioural category: time resting on the water. Time spent in flight and time spent resting are strongly negatively correlated ($R = -0.88$) while time spent foraging is more weakly negatively correlated with time spent in flight ($R = -0.22$), but variance inflation factors (VIFs) are <2.5 , supporting the inclusion of both variables in carry-over effects models (Zuur, Ieno & Elphick 2010; line 216-217). We have now reported these correlations in the text also:

New text at lines 169-171: “Time spent in flight and time spent resting were strongly negatively correlated ($R = -0.88$, $p < 0.001$) while there was a weaker negative correlation between time in flight and time spent foraging ($R = -0.22$, $p < 0.001$).”

Line 181: Again, please tell us here that predictors were not correlated with each other instead of waiting until the end of the section.

As per our response to your previous comment, we have now included these correlations:

New text at lines 169-171: “Time spent in flight and time spent resting were strongly negatively correlated ($R = -0.88$, $p < 0.001$) while there was a weaker negative correlation between time in flight and time spent foraging ($R = -0.22$, $p < 0.001$).”

Lines 184-185: I find it surprising that colony arrival date and laying date are not significantly correlated for females, regardless of what the variance inflation factor is. I would like to see a graph of laying date plotted against arrival date. This might be a supplemental figure.

We have now added a supplemental figure (Appendix D) showing the relationship between colony arrival date and lay date. As the reviewer suggests the relationship is stronger for females than for males, although the variance inflation factors (<2.5) support the inclusion of both variables together in breeding performance models.

Lines 192-194: Put this earlier (see above).

As per our response to your earlier comments, we have now included these correlations:

New text at lines 169-171: “Time spent in flight and time spent resting were strongly negatively correlated ($R = -0.88$, $p < 0.001$) while there was a weaker negative correlation between time in flight and time spent foraging ($R = -0.22$, $p < 0.001$).”

Lines 206-208: I don't understand the nesting rule well enough to evaluate this result. Without the nesting rule, models with boldness only and with sex only are competitive with the “best” model for time in flight. Ignoring those models has possible implications for the carry-over effects models. We hope we have now made clearer the importance of applying the nesting rule in our response to your previous comment (five above) and also in the manuscript (lines 194-199). You are correct that models

containing boldness only and sex only are within 2 AIC units of the “best” (lowest AIC) model, which contains the intercept only. However, as the intercept-only model is nested within the models containing boldness and sex (as the intercept is included in all models), this suggests that they do not add sufficient explanatory power to outcompete the intercept only model.

Lines 211-212: It is not clear to me that this is the correct interpretation. For males, a competing model for arrival date did not include boldness. For both sexes, several competing models for breeding success did not include boldness.

This is correct that some models within the top set did not contain boldness, however, the benefit of the AIC approach is the ability to make inference across all of these models. For ease of this, we perform model averaging for each model set (Table 1) so that average effects across all models can be evaluated. These results indicate effects of boldness on arrival date and offspring survival. However, we have now substantially altered section 3.3 based upon reviewer comments and additional feedback. Previously, we discussed the effects of each fixed effect in turn, but it was pointed out that when these variables are also fitted as interactions, it makes more sense to discuss the effects of interacting terms (if retained) first. We therefore have now restructured this section to discuss models of colony arrival date, lay date, and offspring survival in turn. In the course of this re-structuring, we decided to remove this sentence.

Lines 254-255: Please explain why you think carry-over effects of time flying and foraging were similar. If I understood correctly, effects of foraging activity differed between the sexes and the two activities had opposite effects on females.

By this, we meant that more time spent in flight and more time spent foraging both overall tended to have negative effects on birds (they typically returned later, laid later, and had lower offspring survival). We stated that effects were similar to address our main interpretation of these negative effects (that birds spend more time flying and attempting to forage in order to compensate for poor condition). However, we can see this oversimplified the findings, and we have therefore expanded on this to explain this more carefully:

Amended text at lines 290-293: “In concordance with a number of other studies on seabirds [11,12,14], we detected predominantly negative carry-over effects of time spent both flying and foraging on subsequent breeding performance.”

General comments:

Line 81: In most seabird species, males and females both incubate eggs and rear young, so at least behaviorally, there is little difference in breeding roles. Relative energetic investment in breeding between the sexes is debated.

We can see how it seemed we were saying kittiwakes exhibit substantial sex differences in breeding roles, which is not the case. It perhaps also sounded like we were suggesting the cited study here (by Saino et al. 2017) looks at a species which exhibits strong sex differences in breeding roles, which is also misleading (this study is on barn swallows). We have now amended this line:

Amended text at lines 85-87: “As a result, females may be subject to stronger carry-over effects on breeding, even in species where the sexes do not differ greatly in breeding roles [e.g. 29].”

Lines 298-300: The correct metric here would be lifetime reproductive success. If shy individuals have lower average success than bold individuals, but breed longer, lifetime reproductive success could be equal.

We have altered this to lifetime reproductive success (line 344).

Tables and Figures:

Table 2: It would be helpful to show all models with ΔAIC_c within 2 (those models that were dropped under the nesting rule).

We have now added full model tables to the supplementary materials (Appendix F).

Tables 2 and 3: Please explain what the gray areas mean. The reader can figure it out, but it would be simpler if you explained it.

We have added an explanation of this to the legend of Table 3 (now Table 1, following reviewer 2's suggestions to move Tables 1 and 2 to the supplementary materials).

All figures: I cannot tell the shades of dots apart. Please use more color contrast and larger dots so the colors show more.

We have increased the size of points and used a new colour scale with more contrast in all plots. To make the colour scale clear in all plots, we no longer use separate colours for males and females. Please note that we have also combined the figures into a single figure, as we felt this aids the comparison of all effects simultaneously.

Editorial comments:

Line 22: Delete "of" from "effects of on breeding".

We have amended this.

Line 56 & others: Spell out POLS. We don't need more acronyms that just make it harder for nonspecialists to read the paper.

We have changed this throughout the manuscript.

Line 98: "great control" should perhaps be "greater control" as it is in relationship to males' control.

We have amended this.

Lines 155-158: The second sentence seems to mostly repeat the first sentence. Either combine the two sentences or reword to make it clear how they differ.

We have amended this so that we no longer repeat information twice (lines 195-197).

Lines 212-213: Figure 3 is called out before Figure 2. They should be reversed so Figure 2 is called out first.

This is no longer an issue as we have combined the three figures into one.

Line 242: "high flying activity" is a kind of ambiguous term (sounds like flight altitude). Perhaps "spending more time flying" would be clearer.

We can see how that was ambiguous, and have now changed all previous use of "high flight/flying activity" to "more time spent in flight" or similar throughout the manuscript.

Line 246: Please spell out pace-of-life syndrome. If you don't spell it out in the rest of the discussion, please do so here, because I already forgot what POLS stands for by the time I got to the discussion.

We have changed this throughout the manuscript following your previous comment.

Lines 307-308: Please delete extra words in this sentence.

We have amended this.

Referee: 2

Comments to the Author(s)

The major goal of this research is to test whether individual differences in personality traits (particularly in bold-shy axis) influence the strength of carry-over effects on breeding performance. More specifically, the authors test a hypothesis deriving from pace-of-life syndrome: carry-over effects on breeding performance are weaker in bold (fast-paced) individuals, as they tend to maintain an allocation to reproduction irrespective of previous conditions, while shy (slow-paced) individuals experience stronger carry-over effects because they favor allocation to self-maintenance over current reproduction. They tested this hypothesis in black-legged kittiwakes, which exhibit biparental care

behavior. First of all, they discovered that a negative carry-over effect of non-breeding activity on breeding performance. Second, they further revealed that carry-over effects of non-breeding activity on breeding performance were stronger in shyer individuals: active winters were followed by later breeding onset and worse breeding performance in shy birds, but these effects were weaker or undetected in bolder individuals. Overall, this is a great study and I enjoy reading this manuscript, but there are a few places need to be addressed or further clarified in the article. Following are some major concerns and minor suggestions to the authors:

Thanks very much for your positive and constructive comments!

1. Boldness traits in statistical analysis:

According to the authors' description in the manuscript, they assessed boldness traits using a novel object test in 2017 and 2018, and found these scores are highly repeatable within a single breeding season. Here are some of my concerns:

(1) did the authors only measure boldness in 2017 and 2018 and use this boldness score to fit all the linear mixed-effects models? As far as I read, the non-breeding activity (perhaps also breeding performance) was collected between 2012 and 2018, which means the authors used the boldness data from the later time point to predict the influences on breeding performance in earlier times (2012–2016). My question is that how repeatable this boldness index across different breeding seasons? A challenge when studying pace-of-life syndrome is the fact that behavior traits or physiological traits are highly labile, which means they are sensitive to study methods and even to environmental variation (Beckmann and Biro, 2013; Carter et al., 2013; Biro, Adriaenssens and Sampson, 2014). I understand that personality traits like boldness are considered as stable traits, but they still can vary across at different time points. Use the single-year boldness score (e.g., boldness score in 2017–2018) as an independent variable to test its influence on breeding success across several years (e.g., breeding performance in 2012–2016) might not be an ideal setup.

References mentioned here:

Beckmann C and Biro PA (2013). *Ethology*, 119, 937–947.

Carter AJ, Feeney WE, Marshall HH, Cowlshaw G and Heinsohn R (2013). *Biological Reviews*, 88, 465–475.

Biro PA, Adriaenssens B and Sampson P (2014). *Journal of Animal Ecology*, 83, 1186–1195.

Here we repeat some information included in our earlier response to the editor's related comment. The reviewer is correct that boldness scores were based upon tests conducting in 2017 and 2018 only; meanwhile non-breeding behaviour and breeding performance data were collected over six seasons. However, the repeatability of boldness is assessed across two years, as opposed to within a single year. We are therefore inferring that a behavioural trait stable over 2 years is still representative of birds' behaviour over 5 years (5 years being the longest interval between collection of non-breeding data and first boldness test for any individual). However, we feel that this assumption is valid given that, as noted, boldness is highly repeatable over the 2-year test period ($R \pm S.E. = 0.61 \pm 0.07$), with virtually no reduction compared to that measured in a single year ($R \pm S.E. = 0.68 \pm 0.07$), and in other seabirds boldness has been shown to be repeatable over periods of at least 5 years (Patrick & Weimerskirch 2013, *Ecol. Evol.* 3:4291; Patrick et al. 2017, *J. Anim. Ecol.* 86:1257). Boldness is also widely considered to be repeatable and heritable trait (Dochtermann et al. 2015, *Proc. B.* 282:20142201; see also Patrick & Weimerskirch 2013). While we acknowledge that in an ideal situation, we would have conducted boldness tests over the entire 6-year study period, this study was designed to add personality assays to an existing long-term tracking data archive, which provides a unique opportunity to examine the importance of personality on carry-over effects.

While the reviewer correctly highlights that behavioural traits may be labile, with animals showing plasticity in their personality traits in response to the environment, personality and plasticity are two separate components: personality is the “intercept” value for an individual, which is fixed over time, while plasticity in this trait is the “slope” in response to changes in the environment. As we here quantify repeatable personality variation in standardized conditions, we are not able to quantify or examine plasticity, and we do not expect plasticity to influence birds' personalities over the period of study.

The editor made some very useful suggestions to check for potential confounding effects of the intervals between non-breeding data and personality testing, which we followed and found no evidence of. As the mean interval between non-breeding data and a bird's first boldness test was less than 2 years (1.79), we feel that our results are unlikely to be influenced by the timing of boldness tests. Following your suggestions we have, however, attempted to make this aspect of our data collection clearer to the reader, and outline these justifications in the manuscript:

New text at lines 185-187: "Because boldness tests were conducted exclusively during the final two years of tracking (2017 and 2018), there was an interval of 0-5 years between the collection of non-breeding data and boldness data (mean interval: 1.79)."

(2) The authors mentioned there are 80 individuals were tested in boldness test. Are these the same group of animals sampled for non-breeding activity and scored breeding performance? If yes, how many males and females in this population?

Not all boldness-tested individuals were tracked in their non-breeding activity. However, we included all 80 boldness-tested individuals in the boldness repeatability estimation in order to accurately estimate repeatability from the tested population. Of the 80 individuals tested for personality, 39 were also tracked in their non-breeding activity (for a total of 78 bird-years). 22 of these were males (tracked over 41 bird-years) and 17 were females (tracked over 37 bird-years). We have now made the samples sizes clearer in the text:

New text at lines altered lines 175-177: "We recorded non-breeding activity data over 78 bird-years in total, for 39 boldness-tested individuals over 6 years of study (22 males in 41 bird-years, and 17 females in 37 bird-years), with a mean of two bird-years per individual (range 1-5 years)."

(3) Are all these 5 behaviors representing boldness? Which are the loading weights of each behavior in the PC1? I usually would recommend including a table for principal component analysis output in supplementary data to illustrate the cumulative variance explained in all PCs, just like what the authors did in their earlier publication in 2020 (Harris et al. (2020) *Animal Ecology* 89:68-79). Boldness is represented by PC1 only. We have added the variable loadings and cumulative variance explained to a supplementary table (Appendix B) as the reviewer suggested, which demonstrates the contributions of each variable to PC1.

2. Data interpretation:

The authors indicated that their results revealed predominantly negative effects of winter activity on breeding performance, which I totally agree with it. However, there are some positive carry-over effects on breeding. For example, in males, "flight" activity in the non-breeding season has a positive effect on offspring survival; in females, "forage" activity in non-breeding season also has a positive effect on offspring survival. I think the authors did not further explain these findings. I also think the authors should discuss why these two non-breeding activity traits (forage and flight) sometimes have opposite effects on the same breeding traits. To me, one thing that can be discussed is that, although the authors treated both "forage" and "flight" as the non-breeding activity, they may affect breeding in different ways because the "flight" is an energy-consuming behavior, while "forage" is an energy-refilled behavior. I think that is why we sometimes see the opposite effects of these two behaviors on breeding performance. For instance, in the offspring survival model, forage behavior is positively associated with offspring survival, but flight behavior is negatively associated with offspring survival, though we see both behaviors are negatively correlated with colony arrival date and lay date.

This is an interesting point about the interpretation of effects foraging and flight behaviour. Regarding the interpretation that time spent foraging should have positive effects, we would infer that spending more time foraging only tells us that birds made more foraging *effort*, and not that they necessarily acquired more food; in fact, they may make more foraging effort to compensate for poor foraging efficiency. Accordingly, we expect both time spent in flight and time spent foraging to be energetically costly behaviours. This is the interpretation of other studies which find negative carry-over effects of foraging effort in other seabirds (e.g. Clay *et al.* 2018, *Funct. Ecol.* 32:1832; Daunt *et*

al. 2014, *Ecol.* 95:2077; Shoji *et al.* 2015, *Biol. Lett.* 11:20150671). We have added a line stating this to the methods:

New text at lines 173-175: “We interpret both time spent foraging and time in flight as energetically costly, because we expect minimising the time taken to acquire daily food requirements to be optimal [14].”

We agree that further discussion of the positive carry-over effects of non-breeding activity would be beneficial, and so we have added two new sections to the discussion, firstly addressing these positive effects generally:

New text at lines 305-310: “While we detected exclusively negative carry-over effects of non-breeding activity on breeding phenology, higher offspring survival was predicted by more time spent in flight in males, and more time spent foraging in females. One potential explanation for where non-breeding activity negatively impacted phenology yet positively affected offspring survival is that increased effort can successfully compensate for poor conditions enough to improve chick rearing performance, even if poor conditions results in delayed arrival.”

We then added more to the discussion of how these positive effects differ between males and females, as we feel this is important context for interpreting why these positive effects occur. While the reviewer correctly points out that more time spent foraging is beneficial to beneficial females’ breeding performance, it is still costly to males, therefore we have discussed potential reasons for this here.

New text at lines 390-396: “Firstly, kittiwakes may exhibit sex-dependent non-breeding foraging strategies. Focussing solely on the carry-over effects on offspring survival suggests that that spending more time in flight and less time foraging is beneficial to males, while in females we observed the opposite effect, with spending more time foraging and less time in flight apparently optimal. This pattern could suggest trade-offs between the ability to successfully locate and obtain food, with successful males being less efficient at finding prey but more efficient at capturing it, and the reverse being true for successful females.”

3. Materials & Methods:

The author did not describe how do they assess breeding performance and breeding success in this section. There is no clear definition for some important terms, like “colony arrival date” and “lay date”. Is the date starting from the first day of each year? Or it is starting from the last day of the previous breeding season? Also, what are the relationships of these two traits (colony arrival date and lay date) and breeding performance? I am a bit confused at the beginning because these two traits are negatively correlated with breeding success, but offspring survival is positively correlated with breeding success.

We have added definitions of colony arrival date and lay date to the methods (lines 204-208). Results of carry-over effects models indicate that birds that lay later have poorer breeding performance (i.e. lower offspring survival), and late arrival of females is also associated with poorer performance (Table 1). As these relationships between colony arrival date, lay date and breeding performance are presented in the model results, we did not otherwise present correlations in the methods (whereas following Reviewer 1’s suggestion we have included the correlation between arrival date and lay date at line 215-216/Appendix D, because this correlation has consequence for fitting the two variables together in models of offspring survival). We are unclear about the reviewer’s final question and point about breeding success – offspring survival is our measure of breeding performance, so here the two terms refer to the same variable, rather than two correlated variables. If we have not answered the reviewer’s question here, we would be happy to revisit this if they could clarify?

4. Results and Tables:

I like the way the authors used Akaike’s Information Criterion (AICc) to test and select LMM models! But in the final model (Table 3), I think the authors should also report F-value and p-value for all fixed variables and interaction terms. In the Results section and Tables, there are no p-values

reported in context. I understand p-value is not everything, but without it, it is hard to convince the readers that which variables have significant effects on responding variables. In the figures, since the authors use the LMM regression models, it would be better to report the estimate (slope), R²-values, p-values of the best-fitted lines so that it would be easy for readers to understand how the strong/weak the relationships are.

We understand the appeal of p-values, but to our knowledge, standard procedure is to use either an information-theoretic approach (AIC) or a frequentist approach (p-values), and not both, as this would be redundant (both p-values and AIC are based on the same statistical information). We opted for an information-theoretic approach because of its advantages in multi-model inference, for which reason it is used many other carry-over effects studies (e.g. Clay *et al.* 2018, *Funct. Ecol.* 32:1832-1846; Harrison *et al.* 2013, *PLoS ONE* 8:1-10; Hansen *et al.* 2016, *Conserv. Physiol.* 4:1-10; Sedinger *et al.* 2011, *Am. Nat.* 178:110-123). Table 1 (previously Table 3) currently contains relative variable importance and estimates of effects with standard errors, which should enable readers to soundly evaluate the importance and strength of each effect (and in a way that should be familiar, as many other ecological studies use AIC and not p-values). We feel it would add confusion to add F-values and p-values to our results, but if the reviewers strongly think this would be beneficial, we can include this in the full model tables of the supplementary materials. Estimates (slopes) of effects are already reported in Table 1 (and averaged across all models, in Table 1). We have now also added R²_c and R²_m values (coefficients of determination, equivalent to R² for mixed effects models) to the previous Table 2 as per this suggestion, although please note that we have moved this to the supplementary materials (Table E1, Appendix E), following your later suggestion (second to last). We discuss these coefficients in the results section:

New text at lines 265-268: “Coefficients of determination indicated that the variation explained by fixed effects was between 8-22% for colony arrival date, 8-36% for lay date, and 1-12% for offspring survival (see supplementary materials, Table E1). This suggests that most of the variation in breeding was explained by differences among individuals and years, particularly in offspring survival models.”

Minor comments:

Line 66-69:

If at the population levels, previous study has shown the fast-paced and slow-paced individuals are responding differently to stress in reproductive behavior, what is the advantage of doing similar experiment at individual levels? I think the authors can further elaborate this more to emphasize the importance of their research.

We have now expanded upon this point to highlight the novelty of our approach:

New text at lines 69-76: “While populations of the same species are often shown to vary in pace-of-life, likely driven by their evolution under different ecological conditions [24], empirical examination of the pace-of-life syndrome at the individual level is still relatively scarce, despite theoretical support for its existence [21,25]. An increasing body of evidence demonstrates that individual variation in allocation trade-offs in the form of naturally occurring carry-over effects between seasons, with profound consequences for individual fitness [1]. However, to our knowledge, no study has previously tested whether differences in carry-over effects can be explained by individual variation in pace-of-life.”

Line 95:

But in the present setup, the authors cannot compare the sex differences because (1) they used different model components in males versus in females; (2) they analyze males and females separately.

This is correct that we were unable to analyse differences between the sexes, but control for non-independence of breeding outcomes between paired males and females by fitting models separately by sex. In doing so, we find some patterns that differ between the sexes and we therefore felt it was important to acknowledge this here. We have made a small alteration to this sentence in order to attempt to tone down any suggestion that we make predictions about sex differences to be formally tested here, but we are unsure how else to amend this to more appropriately reflects our approach to

sex differences. If the reviewer thinks it would be best to remove this final sentence, we are happy to do so.

Amended text at lines 100-103: “As kittiwakes are sexually monomorphic and exhibit biparental care [36], we did not expect strong differences between the sexes, but expected that in line with other studies, carry-over effects on the timing of breeding may be stronger in females due to greater control over egg laying.”

Line 110-111:

The authors should also include the protocol of how they conduct blood collecting and DNA extraction for identifying sex in the Methods.

We have now included full sexing methods in Appendix A of the supplementary materials.

Line 159:

How many males and females in these 49 individuals? Do all these individuals have boldness data from each year? Please also see the first point in the major concerns.

We have now inserted this:

New text at lines 175-177: “We recorded non-breeding activity data over 78 bird-years in total, for 39 boldness-tested individuals over 6 years of study (22 males in 41 bird-years, and 17 females in 37 bird-years)”

Next text at line 127: “27 individuals were [boldness] tested in both 2017 and in 2018.”

Line 182-185:

It seems that “arrival date” and “lay date” are associated with one of each other. Is there any multicollinearity issue in this model?

Further to this comment and Reviewer 1’s comment we have added a supplemental plot (Appendix D) showing the relationship between arrival date and lay date. The two variables are positively correlated, although the correlation is not strong, and we determined that we had statistical support to include both variables without encountering multicollinearity issues as variance inflation factors were low (<2.5 ; see Zuur, Ieno & Elphick 2010, *Methods in Ecology and Evolution* 1: 3-14).

Line 210:

I understand that sex variable can be separately analyzed in “colony arrival” and “lay date” models, but “offspring survival” are contributed by both males and females because this species exhibit biparental care (as the authors mentioned in previously), and therefore separately analyzing sex might raise some issues since you wouldn’t be able to count the interactions between the male and female. This is correct that we do not strictly analyse sex differences here due to non-independence of both lay date and breeding success for paired males and females, and also for colony arrival date, partly for continuity, and partly because this would involve fitting three-way interactions between boldness, activity and sex, which we feel would be inappropriately complex for our data. We have therefore removed “sex” from this line to avoid implying that we statistically measure sex effects.

Line 212:

Please also report statistical results (e.g., F-value and p-value) of each result to support the statement. Please also see the fourth point in the major concerns.

Please see our full response to your fourth Major Concern regarding why we do not report F-values and p-values.

Line 213:

Please specify whether it is a positive or negative association.

We have now substantially altered the wording of section 3.3 based upon reviewer comments and additional feedback. Previously, we discussed the effects of each fixed effect in turn, but it was pointed out that when these variables are also fitted as interactions, it makes more sense to discuss the effects of interacting terms (if retained) first. We therefore have now restructured this section to discuss models of colony arrival date, lay date, and offspring survival in turn. This sentence has therefore been removed.

Line 216:

But in both males and females, the authors also find positive carry-on effects on offspring survival. Please also see the second comment in the major concerns. We have now expanded on the findings of positive carry-over effects in the discussion:

New text at lines 305-310: “While we detected exclusively negative carry-over effects of non-breeding activity on breeding phenology, higher offspring survival was predicted by more time spent in flight in males, and more time spent foraging in females. One potential explanation for where non-breeding activity negatively impacted phenology yet positively affected offspring survival is that increased effort can successfully compensate for poor conditions enough to improve chick rearing performance, even if poor conditions results in delayed arrival.”

Line 225-231:

Do all these effects reach statically significant? I understand p-value is not everything, but if it is applicable, the authors should report the statistical results to back up their statement.

Please see our full response to your fourth Major Concern regarding why we do not report F-values and p-values.

Line 244-245:

Not exactly true. In the offspring survival model, the “flight” and “forage” traits have opposite effects on offspring survival.

We have now corrected this sentence so that the statement about negative carry-over effects applies only to interactions:

Altered text at line 281-283: “Interactions between boldness and non-breeding activity supported personality-dependent carry-over effects, and in all supported interactions, we found that negative carry-over effects were stronger in shy individuals than in bolder individuals.”

Line 245:

Any statistical analysis can support this statement?

Our amendment to lines 281-283 (see comment above) now better reflects the meaning of this sentence: we meant that wherever boldness interacted with non-breeding activity to drive breeding, shy individuals experienced stronger negative effects (i.e. colony arrival date for males, lay date for females, offspring survival for males).

Line 287-288:

Which part of data can support this statement?

We have now illustrated this sentence with an example of a finding from our data:

New text at lines 333-335: “For example, shy males arrived earlier to the colony and had higher offspring survival following winters when they spent less time foraging and in flight.”

Line 322-324:

Did the author have “size” data to incorporate into their analysis?

We do have morphometric data for some individuals, but we think incorporating this is beyond the scope of this paper.

Reference:

Please add reference numbers into the reference list. Otherwise, it is difficult to track the citations which are mentioned in the context.

Apologies for this omission, we have amended the reference list to include numbers.

Line 338: “fo 67:1-48”

Is “fo” a typo?

Yes, we have removed this.

Tables:

Did all analysis include “bird identity” and “year” as random variables? If yes, please specify in the table legend.

Yes – we have now updated all table legends to include this.

Table 1 and 2

The “X” and “–” is confusing. The authors can leave the non-included variable as blank.

Also, I would recommend moved Table 1 and 2 to supplementary data.

Moreover, in addition to estimate and importance, the authors should also report F-values and p-values (if applicable) of each variable. Please also see the fourth comment in the major concerns.

We have removed the dashes (-) from tables 1-3 as suggested.

We agree with your recommendation to move Tables 1 and 2 to the supplementary materials, and have done so: please see Appendix C and Appendix E. Please see our response to your earlier comment (point 4 of the Major Concerns) regarding why we do not report F-values and p-values.

Figures:

Please also report slope of the best-fitted line, R²-values, F-values and p-values in either the figure or in the context.

We have now also added R²_c and R²_m values, as we agree this will aid interpretability of model fit, but please see our response to your earlier comment (point 4 of the Major Concerns) regarding why we do not report F-values and p-values.

Appendix B

23-Oct-2020

Dear Dr Harris:

Your manuscript has now been peer reviewed and the reviews have been assessed by an Associate Editor. The reviewers' comments (not including confidential comments to the Editor) and the comments from the Associate Editor are included at the end of this email for your reference. As you will see, the reviewers and the Editors have raised some concerns with your manuscript and we would like to invite you to revise your manuscript to address them.

Research ethics:

Use of animals and field studies:

It is a condition of publication that you make available the data and research materials supporting the results in the article (<https://royalsociety.org/journals/authors/author-guidelines/#data>). Datasets should be deposited in an appropriate publicly available repository and details of the associated accession number, link or DOI to the datasets must be included in the Data Accessibility section of the article (<https://royalsociety.org/journals/ethics-policies/data-sharing-mining/>). Reference(s) to datasets should also be included in the reference list of the article with DOIs (where available).

If you wish to submit your data to Dryad (<http://datadryad.org/>) and have not already done so you can submit your data via this link [http://datadryad.org/submit?journalID=RSPB&manu=\(Document](http://datadryad.org/submit?journalID=RSPB&manu=(Document) not available), which will take you to your unique entry in the Dryad repository.

Please submit a copy of your revised paper within three weeks. If we do not hear from you within this time your manuscript will be rejected. If you are unable to meet this deadline please let us know as soon as possible, as we may be able to grant a short extension.

Best wishes,

Dr Maurine Neiman

Associate Editor Board Member

Comments to Author:

Thank you for addressing the referees comments so thoroughly, and my own. The referees have some remaining queries about the analyses (e.g. personality data distribution, relevance of sex-specific models) and otherwise suggest a few clarifications.

We are very grateful to the editor and the referees for your positive and constructive comments, which we feel have greatly helped to improve the manuscript. We have addressed the referees' further queries below.

We wanted to specifically address the editor regarding Referee 1's request that Appendices A and B be moved to the main text. We have now moved Appendix B (variable loadings from the boldness Principal Component Analysis) to the main paper. However we are at the journal's upper limit of 10 pages, and so are unable to include both Appendix A (details of the molecular sexing protocol) and Appendix B. We feel that the details of the sexing protocol are not essential for readers to interpret the methods and the results, and so prioritised Appendix B when considering these suggestions. If you or the referee do feel it is crucial that Appendix A is included in the main paper, please do let us know,

although this would require that other information in the main paper is shortened or moved to the supplementary materials.

Reviewer(s)' Comments to Author:

Referee: 1

Comments to the Author(s).

I find the manuscript to be improved and close to ready for publication. I think some editing, a little re-wording, and still some better explanations are all that are required. I agree that you should not use both p-values and AIC.

Thank you for your positive and very helpful comments!

Line 55: Why not just say “Among individuals, variation ...” instead of “At the among-individual level, variation ...”?

We have revised this sentence accordingly (line 55).

Line 56: Missing the closing parenthesis.

We have made this correction.

Line 60: I think you mean “reproduction”, not “reproductive”.

We have corrected this.

Line 116: Add “see supplementary material”, which you say for all the other appendices.

We have now added this here.

Lines 134-136: I still think this needs more explanation. Please state exactly what result from the linear model is the single estimate of boldness per individual. You addressed this in your response to reviewers, but did not clarify it in the manuscript.

We have now specifically stated how a single estimate of boldness per individual is extracted from the boldness linear model, using the terminology we used in our previous response to reviewers.

Lines 135-140: “Finally, following [39,40], we fitted a linear model with PC1 as the response variable, and individual ID, test date, breeding stage, and test number as fixed effects. From this linear model we extracted parameter estimates (using the `coef()` function) for each level of the individual ID fixed effect, and used these as a single estimate of boldness per individual. Parameter estimates are regarded as better estimates of individual behaviour than individual point estimates from random effects in mixed models [41].”

Line 158: Delete “for”: “except in cases”.

We have made this correction.

Lines 180-183: Spell out “linear mixed effects models” on line 180 instead of line 183.

We have made this correction.

Line 184: Were bird identity and year nested?

The random effects of bird identity and year were crossed, rather than nested. This is because birds were observed in multiple years, and in each year we also observed multiple birds, such that there was no hierarchical structure to the random effects (one variable was not nested within the other; here we refer to the section on crossed vs. nested random effects in Harrison *et al.* 2018 *PeerJ*, DOI: 10.7717/peerj.4794). For clarity we have now added in the text that our random effects were crossed:

Line 223: “bird identity and year fitted as crossed random effects.”

Line 189: I think the second “interval” should be “interaction”.

We have now corrected this.

Lines 233-238: Refer to Table B1 here. I really think this table should be in the paper, not in the supplementary file. It looks like bold individuals sat tight and very shy individuals flew away. The other behaviors did not contribute much. It would be worth a sentence about the most important “bold” and “shy” behaviors.

Following this suggestion, we have now moved Table B1 to the main paper (now Table 1). We also added a description of the behaviours representing responses interpreted as “bold” and “shy”:

Adjusted text at lines 244-246: “Boldness scores ranged from -0.86 to 1.36 with low values representing when birds left the nest (interpreted as “shy” responses), and high values representing when birds remained sitting on the nest (interpreted as “bold” responses; Table 1).”

Line 392: Delete one “that”.

We have made this correction.

Line 399: Delete the second “than”.

We have made this correction.

Table 1: I think “estimated” in the first line of the caption should be “estimates”. Please re-word the second sentence. It makes no sense. Please refer to Table E1 at the end of the caption.

We have changed “estimated” to “estimates”. We had accidentally included a repetitive line in the second sentence, which we have removed so that the sentence makes sense now:

Amended text in Table 1 legend: “Best supported models were those retained where $\Delta AICc < 2$ and where there was no simpler outranking model”.

We have also added reference to Table E1 (now Table D1) as you suggested, as well as Tables F1-F3 (now Tables E1-E3, the full model output tables).

New text at the end of Table 1 legend: “See supplementary materials Table D1 for summaries of best supported models including coefficients of variation, and Tables E1-E3 for full model outputs.”

Supplementary files:

Appendix A: I would put this in the paper. It is only two paragraphs.

We are currently at the journal’s maximum limit of 10 journals pages, and so have needed to be selective in which information can be included in the main paper. We felt that Appendix B (which you also suggest moving from the supplementary materials below) was more important to include in the main paper than the details of the molecular sexing of the birds, which follows a standardised protocol. We have therefore currently kept Appendix A in the supplementary materials, but please do let us know if you do strongly feel this needs to be moved to the main paper instead of Appendix B.

Appendix B: I think this should be in the paper.

We have now moved this to the main paper (now Table 1).

Appendices C, E, and F: These tables are very helpful.

Thank you, we are glad to hear you found these appendices useful in interpreting the results.

Appendix D: Change “Figure C1” to “Figure D1”.

This was a typo, but as Appendix B is now moved to the main paper, this is now correct.

I’m not sure, but these colors might be problematic for people with some kinds of color blindness. Please check this. Otherwise, I like this figure.

Apologies, we agree our previous colour combination was a poor choice, and have now updated this figure to use a colourblind-friendly palette. Thanks!

Appendix E: You numbered citations in Appendices A and C, but not here. Please be consistent.
We have changed these citations to be in a numbered format.

Appendix G:

Line 2 of the caption: “between 0-5 years” does not work. I think “ranged from 0 to 5 years” would be better.

We have changed the caption accordingly.

You refer to Table 2 of the main paper, but there is only 1 table in the main paper now.

We have corrected this.

Change “Table F1 below” to “Table G1 below”.

This was a typo, but now that Appendix B is moved to the main paper this ordering is correct!

You reversed the estimates for “Boldness x foraging” and “Boldness x flight” in Table 1, but not here.

We have corrected the estimates here now also.

Referee: 2

Comments to the Author(s).

I have carefully read this resubmitted manuscript, and I think this manuscript had great improvements. I also think the authors have thoroughly addressed most of my comments in this revised manuscript. I only have a few additional comments for the authors.

Thank you for the time and consideration you have put into reviewing our manuscript, your comments have been very constructive!

Line 51: change “...by high allocation to current breeding and low survival” to “...by high allocation to current breeding but low survival”.

We have now changed this (now line 56),

Line 164-168: Are there any differences in the length of non-breeding season between shy and bold individuals? I am wondering if the “length of the non-breeding season” would be a better response variable than “colony arrival date”, because it represents how much time an individual spent in preparing for the next breeding season, which can reflect its actual physical condition.

The suggestion that the length of the non-breeding season may reflect birds’ physical condition is an interesting one, but we do not think it would be suitable to replace colony arrival date with length of the breeding season in our carry-over effects models. We are specifically interested in the seasonal interaction between activity during the non-breeding season (the window between colony departure in one year and colony arrival the next) and the subsequent breeding season. Because the length of the non-breeding season is calculated from colony departure date, the model you suggest here would examine the relationship between two metrics from the same season – the duration of the non-breeding season, and birds’ activity during it. Without the between-season component, we feel this would then not constitute a carry-over effect. Further, while other studies have explored carry-over effects on non-breeding behaviour (for example, how breeding success carries over to influence the duration of the subsequent non-breeding season), here we were specifically interested in carry-over effects on breeding, because we predicted such effects to be more strongly influenced by animals’ pace-of-life. To answer your question at the start of this comment, we did look into whether the length of the non-breeding season differed by boldness, but find no suggestion of a relationship (see figure below), and so at present do not feel this ought to be included in the main paper.

Line 182-183 and other models: Is there any transformation required for the variables to meet the normality assumption in LMMs? If yes, please specify these variables and describe how they are transformed.

Boldness was negatively skewed, and so we reflected and then square-root transformed boldness, before reflecting it back to the original direction. This is specified further on in the paragraph:

Line 231-232: “Boldness was reflected and square-root transformed to adjust for negative skewness, and then reflected back to the original direction, to meet normality assumptions.”

No other transformations were required.

Line 198: Any reference for setting up this criterion (i.e., < 2 AIC units)?

We have now added a reference to Burnham & Anderson (2001; see page 114) to support our use of a Δ AIC cut off of 2 units here (line 202)

Line 217: please specify what the statistic test is here.

We have now more clearly specified that the value of < 2.5 refers to variance inflation factors as a statistic, and moved the citation to make clearer where we sourced this cut-off.

Lines 219-221: “for all models we inspected variance inflation factors (VIFs) of predictor variables and found no evidence of collinearity (VIFs < 2.5 in all cases, indicating minimal collinearity [55])”

Line 229: I think this should be Appendix G, not F.

This was a typo, but following a suggestion from another referee to move Appendix B to the main paper this is now correct.

Line 244: There are no Table D1 in the Appendix. Do the authors mean Table C1 here?

This was a typo and should have referred to Appendix F of the supplementary materials (the full model tables). We have changed this now.

Line 248-249, Line 314 and Figure 1: In this paragraph and several other places, the authors mentioned that negative carry-over effects were strongest in shy individuals, but how do they determine the effect is strong or weak? I cannot tell the level of significance between shy and bold individuals from Table 1, and there were no statistical values reported in Figure 1. In Figure 1g, it seems that there are negative carry-over effects in both shy and bold individuals, did the authors determine the strength of carry-over effects by slopes or R-square value?

We do not infer differences in the strength of carry-over effects between bold and shy birds from R^2 values or another test statistic, because boldness is fitted as a single, continuous variable, and grouped only for plotting purposes. Therefore, when discussing the strength of carry-over effects, we are referring to the slopes of the relationship between non-breeding activity and breeding response, as you suggest: steeper slopes are interpreted as stronger effects. We have now clarified that we are referring to the steepness of the slopes, and directly refer the reader to the figure here:

Amended text at lines 256-258: “Among males, steeper slopes between arrival date and time spent foraging and in flight (Figure 1a-b) indicate negative carry-over effects were strongest in shy individuals.”

For added clarity, we also have now specifically stated in the main text that boldness was fitted as a continuous measure in all analyses:

New text at lines 221-222: “Boldness was fitted as a continuous measure in all analyses, and was grouped in figures for illustrative purposes only.”

Also, in Figure 1, the authors mentioned that estimates are presented for the boldest individuals (+1 standard deviation) and shyest individual (-1 standard deviation); I wonder how many individuals are categorized in boldest and shyest groups? Can they represent most of the bold and shy individuals in the population? I am worried about if the results might be driven by few individuals with extremely behavioral phenotypes (e.g., extremely bold or shy individuals).

Here again, we emphasise that boldness is fitted as a continuous variable in all models, and we only group individuals by boldness in order to aid visualisation. These groups therefore do not have bearing on the statistical results. As mentioned above, we appreciate this may be confusing, and so have now specifically stated that boldness is fitted as a continuous measure in the main text, and also edited the legend of Figure 1 to make this clearer.

New text at lines 221-222: “Boldness was fitted as a continuous measure in all analyses, and was grouped in figures for illustrative purposes only.”

Amended text in the Figure 1 legend: “Boldness is fitted as a continuous measure in all analyses. For plotting purposes only, where an interaction between boldness and activity was supported, estimates are presented for the boldest individuals (+1 standard deviation from the mean) in purple solid lines, and for the shyest individuals (-1 standard deviation from the mean) in green dashed lines.”

Line 253-255: I am wondering if "lay date models" are important in males because the lay date is majorly controlled by the female itself, and therefore this trait may not be directly related to male reproduction. Do you think this might be the reason why the authors did not see the effects of the interaction term (boldness x activities) on these traits? If the authors used the behavioral traits which is more relevant to male reproduction (e.g., first courting date or first copulation date) in these models, would they expect to see a different result?

This is a very interesting point. We had expected that if lay date is predominantly controlled by females, we would see carry-over effects on lay date among females only, and we do discuss this in the manuscript already:

Lines 100-104: “As kittiwakes are sexually monomorphic and exhibit biparental care [35], we did not expect strong differences between the sexes, but expected that in line with other studies, carry-over effects on the timing of breeding may be stronger in females due to greater control over egg laying.”

Lines 383-385: “A number of studies in birds have reported that carry-over effects on the timing of egg laying are stronger in females than in males [26–29], attributing this to female control over the timing of egg laying [68].”

But here, we think the reviewer is suggesting that predominant control over egg laying may be the underlying reason that an *interaction* between boldness and carry-over effects on lay date is observed among females only. Indeed, while male condition may also influence the timing of breeding behaviour which in turn drive the timing of laying, females may have greater control over lay date, allowing them to adjust lay date to their personality. We think this is an interesting extension of our discussion of sex-specific carry-over effects, and have adjusted the relevant discussion paragraph to reflect this:

Amended text at lines 385-392: “Here, we found that in kittiwakes, the timing of laying was related to the non-breeding activity of both sexes. This implies that the timing of laying is driven by both female and male condition: males in better body condition may advance their partner’s lay date through earlier engagement in breeding behaviours such as nest building, courtship feeding and, ultimately, copulation [69]. However, only among females did we detect an interaction between boldness and non-breeding activity on lay date. While male condition may influence the timing of breeding activities, the interaction between boldness and non-breeding activity among females may suggest that females are better able to optimise the timing of laying to their pace-of-life.”

Line 388-390: Please add references to support this sentence.

We have now added references to support our suggestion that behavioural and/or physiological differences between males and females may drive differences in carry-over effects:

Added references at lines 398-400: “This sex difference in the non-breeding behaviours driving carry-over effects may be the result of a number of behavioural and physiological inequalities between males and females [27,28].”

Line 399: remove extra “than” in this sentence.

We have made this correction.

Line 400: I am still interested to see if the authors pool male and female as one "sex" variable and include it in the “offspring survival model”, would they see a different result? The offspring survival should account for both males' and females' investment. Therefore, analyzing males and females separately might raise some issues since they are unable to count the interactions between both sexes. I understand the three-way interactions (sex, boldness, and activity) may make this model over complex. Still, it may not necessarily need to be included in the models depending on its importance. We still feel it would be incorrect to pool males and females into a single offspring survival model and include a “sex” variable in the model, for two main reasons. Firstly, this would mean duplicating samples to fit the same nest outcome for both the male and the female from each nest. This would amount to pseudoreplication of offspring survival samples, and impede interpretation of the model’s results. This is a common issue for data of this type, whereby the same outcome is shared by two individuals of different sexes, and is frequently handled by fitting separate models for each sex as we do. For some examples, please see: Bize *et al.* (2005), *Funct. Ecol.* 19:405; Ouyang *et al.* (2011), *Proc. Roy. Soc. B.* 278:2537; Patrick *et al.* (2015), *Proc. Roy. Soc. B.* 282:20141649; Morinay *et al.* (2018) *Front. Ecol. Evol.* 5:1.

Secondly, if (despite the pseudoreplication issue) we were to fit a pooled model, the reviewer suggests that three-way interactions may be unnecessary in a pooled model, and that fitting just the fixed effect of sex may be sufficient to capture the sex differences in carry-over effects. However, we find differences between the sexes in the two-way interactions between boldness x activity (Figure 1 of the main paper). Therefore, failure to fit a three-way interaction here would mask these results, preventing the sex-specific two-way interactions from being detected. Fitting sex as a non-interacting term would not account for this, but would only fit an effect of sex on offspring survival, which we feel is biologically uninterpretable in this system.

We do understand the reviewer’s interest in understanding how both male and female carry-over effects collectively contribute to the survival of offspring, and we think this would make an extremely interesting follow up, exploring carry-over effects and parental conflict/cooperation. However, given the complexity of the models that would be required to conduct this analysis alongside exploring the contributions of male and female personalities, we feel this is beyond the scope of our dataset and of the current study. Ultimately, we feel the use of separate models for males and females is valid and appropriate for the questions we address here, and that models pooling the sexes would not be a valid approach.